# Loss of vascular endothelial notch signaling promotes spontaneous formation of tertiary lymphoid structures

Susanne Fleig [1,2,13,14], Tamar Kapanadze[1,2,14], Jeremiah Bernier-Latmani[3], Julia K. Lill[4], Tania Wyss [3,5], Jaba Gamrekelashvili [1,2], Dustin Kijas[1,2], Bin Liu [6], Anne M. Hüsing [2], Esther Bovay[7], Adan Chari Jirmo [6,8], Stephan Halle [9], Melanie Ricke-Hoch [10], Ralf H. Adams [7], Daniel R. Engel [4], Sibylle von Vietinghoff[2,11], Reinhold Förster [9], Denise Hilfiker-Kleiner[10,12], Hermann Haller[2], Tatiana V. Petrova [3] & Florian P. Limbourg [1,2✉]

Tertiary lymphoid structures (TLS) are lymph node-like immune cell clusters that emerge during chronic inflammation in non-lymphoid organs like the kidney, but their origin remains not well understood. Here we show, using conditional deletion strategies of the canonical Notch signaling mediator *Rbpj*, that loss of endothelial Notch signaling in adult mice induces the spontaneous formation of bona fide TLS in the kidney, liver and lung, based on molecular, cellular and structural criteria. These TLS form in a stereotypical manner around parenchymal arteries, while secondary lymphoid structures remained largely unchanged. This effect is mediated by endothelium of blood vessels, but not lymphatics, since a lymphatic endothelial-specific targeting strategy did not result in TLS formation, and involves loss of arterial specification and concomitant acquisition of a high endothelial cell phenotype, as shown by transcriptional analysis of kidney endothelial cells. This indicates a so far unrecognized role for vascular endothelial cells and Notch signaling in TLS initiation.

[1] Vascular Medicine Research, Hannover Medical School, 30625 Hannover, Germany. [2] Department of Nephrology and Hypertension, Hannover Medical School, 30625 Hannover, Germany. [3] Vascular and Tumor Biology Laboratory, Department of Oncology UNIL CHUV and Ludwig Institute for Cancer Research, Lausanne, Switzerland. [4] Department of Immunodynamics, Institute for Experimental Immunology and Imaging, Medical Research Centre, University Hospital Essen, 45147 Essen, Germany. [5] SIB Swiss Institute of Bioinformatics, Lausanne 1015, Switzerland. [6] Hannover Medical School, Biomedical Research in Endstage and Obstructive Lung Disease (BREATH), Member of the German Center for Lung Research (DZL), Hannover, Germany. [7] Max-Planck-Institute for Molecular Biomedicine, 48149 Muenster, Germany. [8] Department of Pediatric Pneumology, Allergology and Neonatology, Hannover Medical School, Hannover, Germany. [9] Institute of Immunology, Hannover Medical School, 30625 Hannover, Germany. [10] Department of Cardiology and Angiology, Hannover Medical School, 30625 Hannover, Germany. [11] Division of Medicine I, Nephrology section, UKB Bonn University Hospital, Bonn, Germany. [12] Department of Cardiovascular Complications of Oncologic Therapies, Medical Faculty of the Philipps University Marburg, 35037 Marburg, Germany. [13]Present address: Department of Geriatric Medicine (Medical Clinic VI), RWTH Aachen University Hospital, 52074 Aachen, Germany. [14]These authors contributed equally: Susanne Fleig, Tamar Kapanadze. ✉email: limbourg.florian@mh-hannover.de

Tertiary lymphoid structures (TLSs) are de novo generated lymphoid structures in non-lymphoid organs like the kidney that develop in response to chronic inflammation and sustain chronic immune responses[1,2]. TLSs develop in a variety of chronic inflammatory lesions[3] and occur in auto-immune kidney diseases like Lupus nephritis[4], ANCA-associated glomerulonephritis[5], membranous glomerulonephritis[6] or IgA-Nephritis[7], but also in kidney transplants[8]. During infection, TLSs are beneficial and are associated with pathogen clearance and increased survival. However, TLS can destroy normal kidney tissue and exacerbate autoimmune diseases and chronic rejection, suggesting that TLS are therapeutic targets in these conditions[1]. TLS resemble lymph nodes in cell composition and structure and form functional germinal centers[9]. Lymphocytes are attracted by chemokines like CXCL13 (B cells) and CCL19 (T cells) secreted by local stroma, and they in turn secrete lymphotoxins, which promote stroma differentiation towards lymphoid tissue fibro-blastic reticular cells (FRC) and follicular dendritic cells (FDC)[10].

While many studies have focused on immune cell contribution, little is known about the vascular regulation of TLS generation. TLSs contain specialized high endothelial cells (HECs), often organized in high endothelial venules (HEV), that recruit passing myeloid cells and lymphocytes via peripheral lymph node addressin PNAd, a glycoprotein ligand for L-selectin (CD62L) expressed by high endothelial venules in lymph nodes required for lymphocyte egress[9]. HECs strongly differ from lymph node capillary endothelial cells (ECs) in transcriptional signature[11–13]. HECs demonstrate enrichment in transcripts involved in the regulation of inflammatory response, leukocyte migration, and lymph node development. Interestingly, Notch signaling com-ponents and its downstream targets, as well as endothelial cell differentiation markers, are strongly downregulated in HEC[11].

Canonical Notch signaling is an evolutionary conserved, cell-contact dependent signaling pathway[14]. Activation of one of four membrane-bound Notch receptors by Notch ligands leads to Notch receptor intracellular domain (NICD) cleavage and translocation to the nucleus, where it associates with DNA-bound Rbpj and initiates transcription of target genes. Notch is a key player in vasculo- and angiogenesis during development[14–16], and regulates arterial phenotype of endothelial cells and arterial EC identity in the adult[17,18]. Inversely, suppression of Notch by COUP-TF2 permits venous endothelial phenotype[19].

Here we show that conditional loss of Notch signaling by deletion of the canonical mediator Rbpj in blood vascular-endothelial cells modifies arterial endothelial identity and shifts it to an HEV-like phenotype, which in turn is associated with the spontaneous formation of TLS in mouse kidney, liver, and lung. Our results highlight an essential and unexpected role of arterial endothelium in formation of TLS and suggest targeting the endothelial Notch pathway as a novel approach for modulating organ-specific immunity.

## Results and discussion
### Loss of endothelial Notch signaling induces spontaneous for-mation of TLSs.
To study the role of Notch signaling in vascular and immune homeostasis in the kidney, we generated $Cdh5Cre^{ERT2};Rbpj^{fl/fl}$ ($Rbpj^{\Delta EC}$) transgenic mice by crossing conditional alleles of the canonical Notch effector Rbpj and an endothelial-specific and inducible Cre-recombinase[20,21]. We then induced Cre-recombinase activity by Tamoxifen injections at 7–9 weeks of age[22], after completion of developmental angio-genesis and vascular remodeling (Fig. 1A), and confirmed recombination of the Rbpj locus and downregulation of the Notch target gene Hey1 (Supplementary Fig. 1A, B).

Twelve weeks after induction, $Rbpj^{\Delta EC}$ showed a higher frequency of mature B-lymphocytes (CD45+, CD19+, B220+) and T-lymphocytes (CD45+, CD3+) in the kidney by flow cytometry, while B cell frequencies were reduced in peripheral blood and bone marrow (Fig. 1B, Supplementary Fig. 2A, B; 1L). The increase in renal B cells was driven by follicular B lymphocytes, not progenitors or other subtypes (Fig. 1C; Supplementary Fig. 1C, D). No changes were detected in B cell bone marrow niche cytokine expression of Il7 and Cxcl12 (Supplementary Fig. 1F). Furthermore, cell numbers and frequencies of B and T cells in secondary lymphoid organs such as the spleen or lymph nodes were comparable between $Rbpj^{\Delta EC}$ and control mice (Fig. 1B), and spleen size was not changed (Supplementary Fig. 1G). $Rbpj^{\Delta EC}$ mice also showed an increased frequency of dendritic cells in the kidney, bone marrow and spleen (Fig. 1B, lower panel; Supplementary fig. 1H). In contrast, we observed no difference in neutrophilic granulocytes, monocyte subsets or macrophages between $Rbpj^{\Delta EC}$ and control mice, and no signs of overt systemic inflammation (Fig. 1B, lower panel, supplementary Fig. 1H). Together, these findings suggest active lymphocyte recruitment to the kidney.

Histologically, we observed significant, but localized lympho-cytic infiltrations, consistent with TLSs[23], clustered around segmental and interlobar arteries in kidneys of $Rbpj^{\Delta EC}$ mice, but not in littermate controls (Fig. 1D, E). Aside from these infiltrations, the overall kidney architecture was preserved, although TLS displaced normal renal structures (Fig. 2A, Supplementary Fig. 2A). In areas of lymphocyte infiltration, Masson Trichrome and Sirius Red staining demonstrated focal interstitial matrix deposition and areas of fibrosis (Supplementary Fig. 2A). Overall kidney function as measured by serum creatinine and proteinuria was not altered in $Rbpj^{\Delta EC}$ mice; however, kidney Havcr1 gene expression (kidney injury molecule 1) was significantly increased, suggesting subclinical renal injury (Supplementary Fig. 2B–D). Interestingly, 3D reconstruction of light sheet microscopy images of CD31- and B220-stained whole kidneys confirmed extensive but stereotyped periarterial expan-sion of TLS along the segmental and interlobar arteries, forming central conglomerates in the renal medulla (Fig. 1E, Supplemen-tary Movie 1). Thus, loss of endothelial Notch signaling induced spontaneous formation of renal TLS around second and third-order arteries without overt systemic or chronic inflammation.

Since $Rbpj^{\Delta EC}$ mice have a pan-endothelial deletion of Notch signaling, we next analyzed liver and lung, parenchymatous organs prone to develop TLS[24–28], and also the heart[29]. Consistent with a general role of Notch signaling in the regulation of parenchymatous TLS, we found regular TLS in all livers and lungs analyzed, but not the heart (Fig. 1G, Supplementary Fig. 2E–G).

**TLSs show regular lymphoid and stromal tissue architecture and germinal center formation**. To characterize in more detail the TLS that formed spontaneously in $Rbpj^{\Delta EC}$ mice, we stained kidney cross-sections with markers of proto-typical lymphoid tissue cell components. As a general principle, TLSs were orga-nized around a central artery and were structured by a network of stromal cells expressing the FRC marker podoplanin[30]. This network was connected to the central artery and showed occa-sional enlarged and filled lymph vessels (Fig. 2A).

TLS contained B and T cells organized into distinct T and B cell-zones (Fig. 2B), and staining for GL7- and IgD-positive B cells showed regular germinal center formation (Fig. 2C)[31], which was in line with a follicular B cell phenotype by flow cytometry (Fig. 1C). Furthermore, a high percentage of B cells were KI67+,

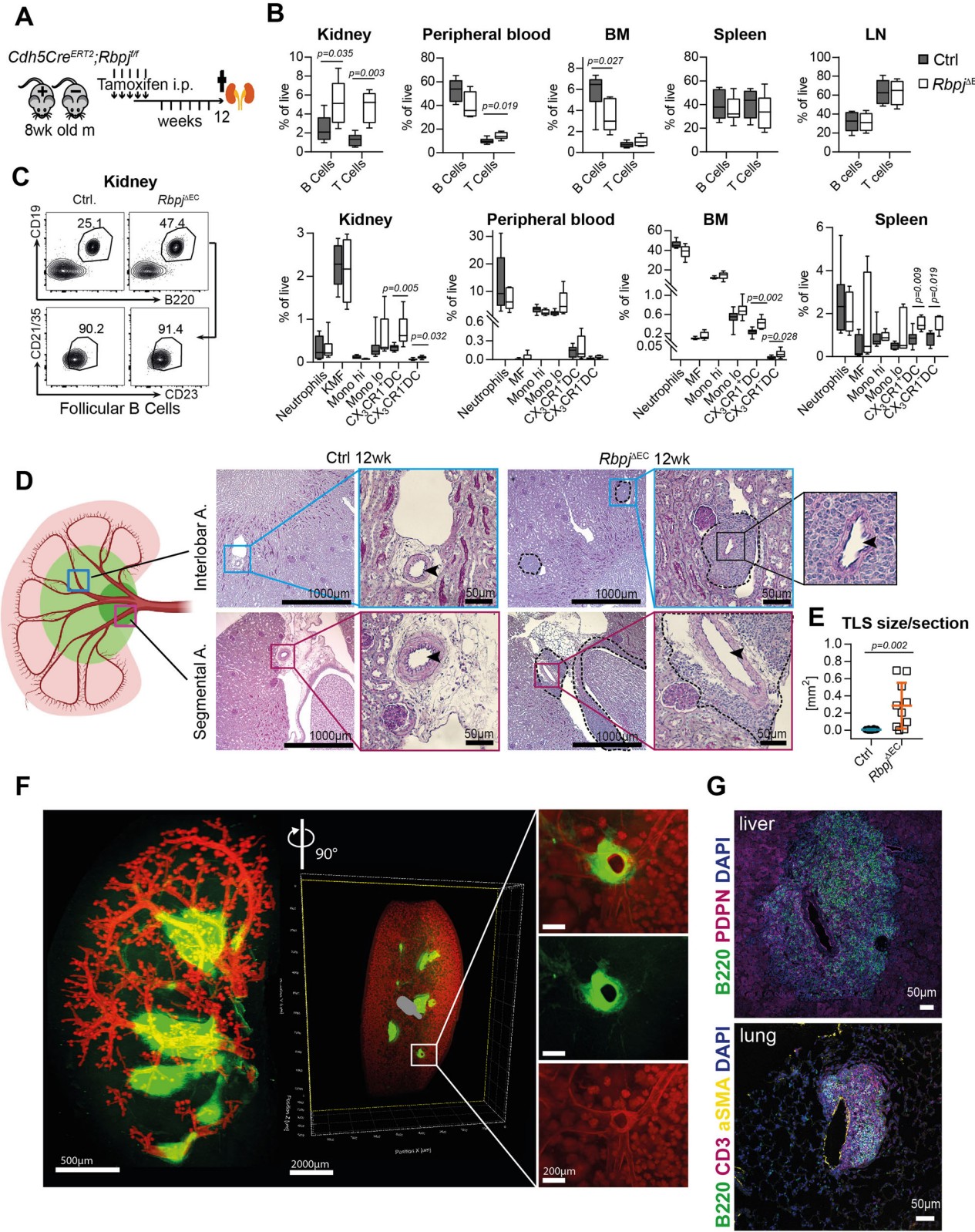

indicating active proliferation (Fig. 2F, I). Thus, these data demonstrate a B cell phenotype consistent with mature TLS.

We next studied the stromal components in TLS. CXCL13 is the major chemokine expressed by FDC attracting B-lymphocytes during the formation of lymphoid structures[30,32]. B cells in *Rbpj^ΔEC* kidneys clustered around CXCL13-expressing cells (Fig. 2D), which co-expressed CD21/35, consistent with FDC phenotype[28,33]

(Fig. 2E). In spleen or lymph nodes, stromal fibroblastic reticular cells (FRC) form conduits that guide B and T cells[30]. Stroma cells in *Rbpj^ΔEC* TLS stained positive for FRC-markers podoplanin (PDPN, Fig. 2A) and ER-TR7 (Fig. 2G)[1,10,34–36], forming conduits distinct from PDPN+/Lyve1+ lymphatic vessels (Fig. 2A, insets). Notably, the conduit network formed by FRC in TLS was centered around and closely attached to the central artery, extending in a

**Fig. 1 Spontaneous periarterial formation of TLS in conditional endothelial *Rbpj* mutant mice. A** Induction protocol for *Cdh5Cre*[ERT2]; *Rbpj*[fl/fl] mice and Cre-negative littermates. **B** Flow cytometry, % of live cells, box plots with mean, 25–75th percentile (Inter-Quartile-Range, IQR, bounds of box) and total range (min-max, whiskers); Mann–Whitney test, two-tailed. CTRL n = 9, KO n = 6 mice; 3 independent experiments. **C** Flow cytometry representative plots for % follicular B lymphocytes in Ctrl and *Rbpj*[ΔEC] kidneys. Numbers indicate % of live CD45 + (upper panel)/% of CD45+/CD19+ cells (lower panel). **D** PAS staining of representative paraffin-embedded kidney sections in different arterial segments (sketch on left created with biorender.com). Upper row, interlobar arteries, lower row segmental arteries, magnification ×50 (overview, bar = 1000 μm) and boxed details magnification ×200 (bar = 50 μm). Inset: arterial lumen (arrowhead) within TLS structure. Experiment independently repeated with similar results >×3. **E** Quantification of infiltrated area [in mm²] per transversal kidney cross-section (sum of all infiltrated areas per section). N = 10 mice per group, Mann–Whitney test, two-tailed, exact *p*-value 0.0021; Graph: Scatter dot blot, mean, standard deviation (SD). \*\**p* < 0.01. **F** Whole-mount kidney staining and light sheet imaging of CD31 (red) and B220 (green) of *Rbpj*[ΔEC] kidney; 3D reconstruction with IMARIS software; ventral view left with a filter for larger vascular structures; sagittal view middle and right, with magnification of inset. Representative image, kidneys of n = 3 mice were stained. **G** Representative images of IF stained liver (upper) and lung (lower image) TLS in *Rbpj*[ΔEC] mice, B220 positive cells in green. Organs of n = 3 mice were stained with similar results. Scale bars as marked. Source data are provided as a Source Data file.

honeycomb pattern into the periphery (Fig. 2G). Furthermore, immunostaining also revealed expression of the TNF-superfamily member RANKL in the periphery of TLS, which is expressed by lymphoid tissue inducer cells and mesenchymal lymphoid tissue organizer (LTO) cells[37] and induces B cell chemokines in FRC[38] (Fig. 2H).

To corroborate a TLS molecular signature, we analyzed gene expression of whole kidney samples from control or *Rbpj*[ΔEC] mutant mice, which revealed significant upregulation of *Cxcl13, Cxcr5, Cxcr4, Ccl19,* and *Baff*, prototypical genes involved in attraction and accumulation of B cells, and to some extent T cells, in TLS (Fig. 2J). This extended analysis demonstrates on a molecular, cellular, and structural level the formation of TLS after induced loss of function of endothelial Notch signaling in adult mice.

**TLS formation occurs independent of cardiac or lymphatic disease phenotypes but involves arteries as general guiding structures.** To address the role of potential confounding explanations for the development of TLS we performed several control experiments. *Rbpj*[ΔEC] mice develop cardiac failure around 13–16 weeks of induction (Supplementary Fig. 2I)[29]. In order to investigate whether renal TLS formation in *Rbpj*[ΔEC] mice was secondary to cardiac failure we examined a genetically different mouse model of heart failure induced by myocardial-restricted deletion of Stat3 (*αMHCCre;Stat3*[fl/fl] = *Stat3*[ΔMyoc])[39]. Male *Stat3*[ΔMyoc] mice develop age-related heart failure with dilatative cardiomyopathy (DCM) associated with lower myocardial capillary density starting at 6 months of age (Fig. 3A)[39]. However, although mice developed the full clinical picture of heart failure, indicated by an increased heart weight and expression of *Nppa* (ANP) (Fig. 3B), there was no evidence for TLS formation in the kidney of mutant mice, neither by lymphocyte quantification by flow cytometry (Fig. 3C) nor histologic examination (Fig. 3D). Furthermore, gene expression profiling also did not show TLS-associated changes observed in *Rbpj*[ΔEC] mutant mice (no difference in kidney mRNA for *Cxcl13, Cxcl12, Cxcr5, Cxcr4,* and *Ccl19*, supplementary fig 3A. Thus, heart failure per se does not lead to spontaneous development of TLS.

Our genetic targeting strategy also affects lymphatic endothelial cells[40]. Since TLS in *Rbpj*[ΔEC] mice showed lymph vessels filled with mononuclear cells (see Fig. 2A, purple arrowheads), which might be caused by obliteration of efferent lymph vessels as described in the setting of CLEC2-deficient lungs[41], we first stained whole kidneys with antibodies to B220 and LYVE1/Prox1 and imaged TLS/lymphatic architecture by light sheet microscopy. After 3D reconstruction, efferent lymphatic vessels appeared open with a continuous lumen within and outside the TLS structure (Fig. 3E, Supplementary movie 2), resembling normal renal lymphatics[42].

To test whether TLS formation was secondary to lymphatic-EC targeting of Notch signaling, we conditionally deleted *Rbpj* with the lymphatic-EC specific[43], inducible *Prox1Cre*[ERT2];*Rbpj*[fl/fl] mouse model (*Rbpj*[ΔLEC], Fig. 3F; recombination control in Supplementary Fig. 3B, C). In contrast to *Rbpj*[ΔEC] mice, which showed lower body weight at the end of the observation period (Supplementary Fig. 2H), age-matched *Rbpj*[ΔLEC] mice had normal weight compared to littermate controls and also showed no cardiac phenotype (Fig. 3G). Importantly, there was no evidence of TLS development in *Rbpj*[ΔLEC] mice, neither by histologic examination of the periarterial area covered by lymphatic/mononuclear cells (Fig. 3H), nor by immunofluorescent staining and quantification of B-/T-lymphocytes (Fig. 3I). Thus, induced Notch loss of function in lymphatic EC is not sufficient to induce spontaneous TLS formation. Together, these data demonstrate that loss of function of vascular-endothelial Notch signaling promotes TLS formation.

To examine whether arteries are common guiding structures for TLS formation in kidney we analyzed an unrelated, inflammatory mouse model of unilateral ischemia-reperfusion injury (Supplementary Fig. 4A–C). Eleven weeks after injury, TLSs were found around segmental arteries in a stereotyped manner (Fig. 3K, Suppl. Figure 4D), which corroborates previous observations in several unrelated injury models, but also human kidneys[10]. This suggests periarterial organization as a general principle for TLS formation.

**Loss of endothelial Notch signaling induces an EC phenotype shift from arterial to high endothelial signature and is associated with chronic inflammatory kidney disease.** Notch is a key regulator of endothelial identity. Active Notch signaling induces and maintains an arterial phenotype[15,44], while low Notch signaling is associated with a venous phenotype, including the HEC phenotype found in lymph nodes[11]. We therefore hypothesized that induced endothelial Notch loss-of-function would lead to arterial dedifferentiation and an HEC phenotype shift, thereby promoting TLS formation.

Compared to kidneys of littermate controls, kidneys of *Rbpj*[ΔEC] mice showed significantly reduced expression of the arterial marker *Efnb2*, involved in the arterial specification, but upregulated expression of *Aplnr*, identified as a venous marker which is downregulated at the onset of arterial specification[45] (Fig. 4A).

Furthermore, in addition to reduced expression of the Notch target gene *Hey1* (Supplementary Fig. 1A), the Notch signaling components *Notch1*, *Dll4* and *Jag1*, which are enriched in arterial endothelium[45,46], were also downregulated in *Rbpj*[ΔEC] kidneys, consistent with loss of arterial EC identity (Fig. 4A).

At the same time, expression of peripheral lymph node addressin PNAd, glycoprotein ligands expressed by high

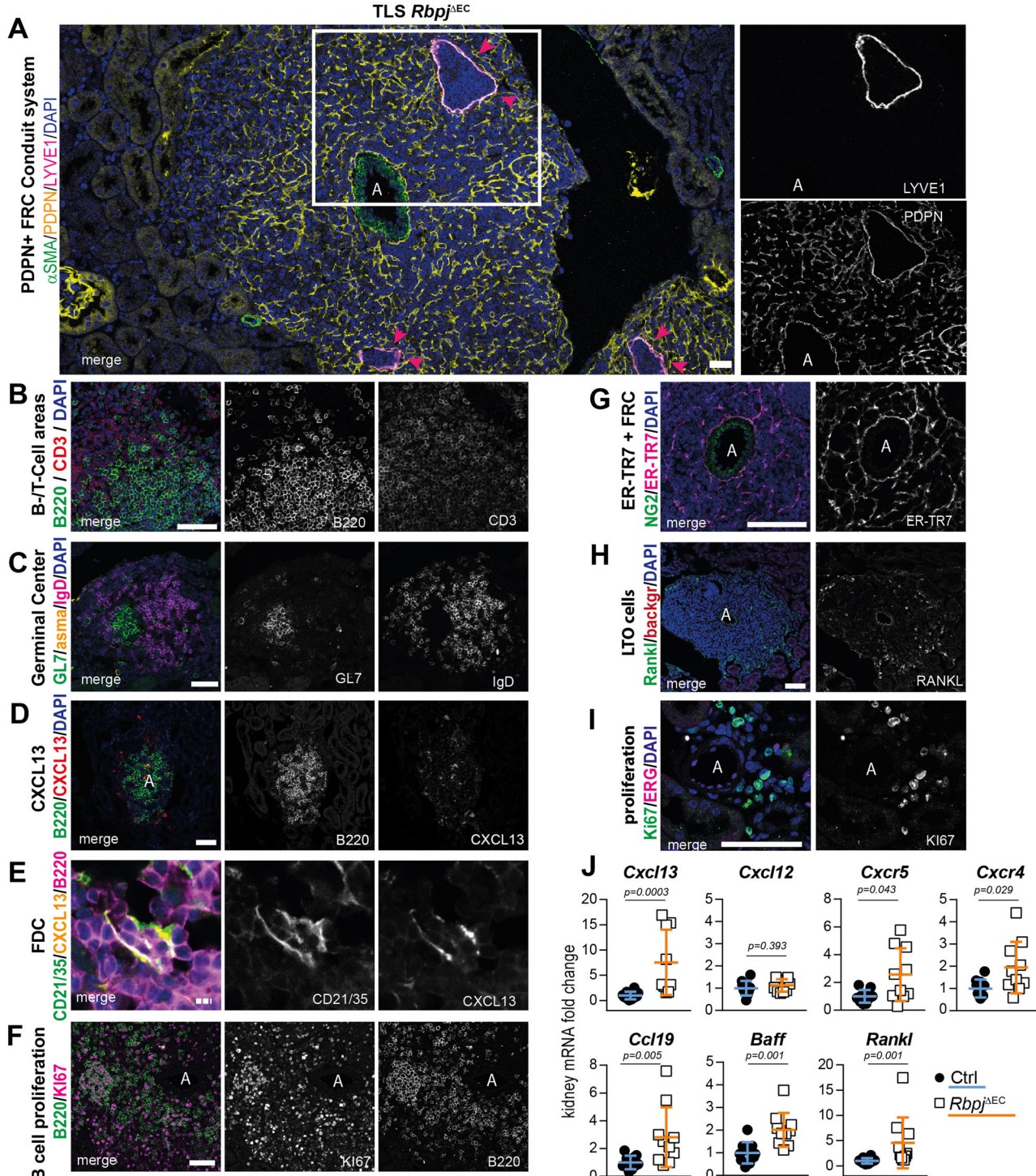

**Fig. 2 Molecular, cellular and structural composition of periarterial TLS. A–I** Immunofluorescence staining and confocal laser scanning microscopy of representative *Rbpj*^ΔEC kidney samples, merged and single channels as indicated. "A" indicates artery. Optical Magnification 200x; different scan areas (see scale bars). Scale: solid bar 50 µm, dotted bar 10 µm (2E). Each micrograph is representative of at least 4 biological replicates. **J** Whole kidney mRNA expression, relative fold change to control gene *Rps9*, n = 10/group. Graphs: Scatter dot blot, mean, SD (whiskers). Mann–Whitney test, two-tailed, Exact p-values: *Cxcl13*, p = 0.0003; *Cxcl12*, p = 0.393; *Cxcr5*, p = 0.0433; *Cxcr4*, p = 0.0288; *Ccl19*, p = 0.0052; *Baff*, p = 0.0007; *Rankl*, p = 0.0005. Source data are provided as a Source Data file.

endothelial venules in lymph nodes required for lymphocyte egress via L-selectin (CD62L)[47], was markedly enhanced in selected vascular beds in mutant kidneys. Remarkably, PNAd staining revealed strong expression in arterial endothelium in central TLS arteries, while endothelium in control kidneys showed no evidence of PNAd expression (Fig. 4B; Supplementary Fig. 4E). Furthermore, PNAd decorated the apical aspects of arterial endothelium, as observed in differentiated HEC in lymph nodes. In addition, peritubular capillaries and vasa recta in the papilla also expressed PNAd.

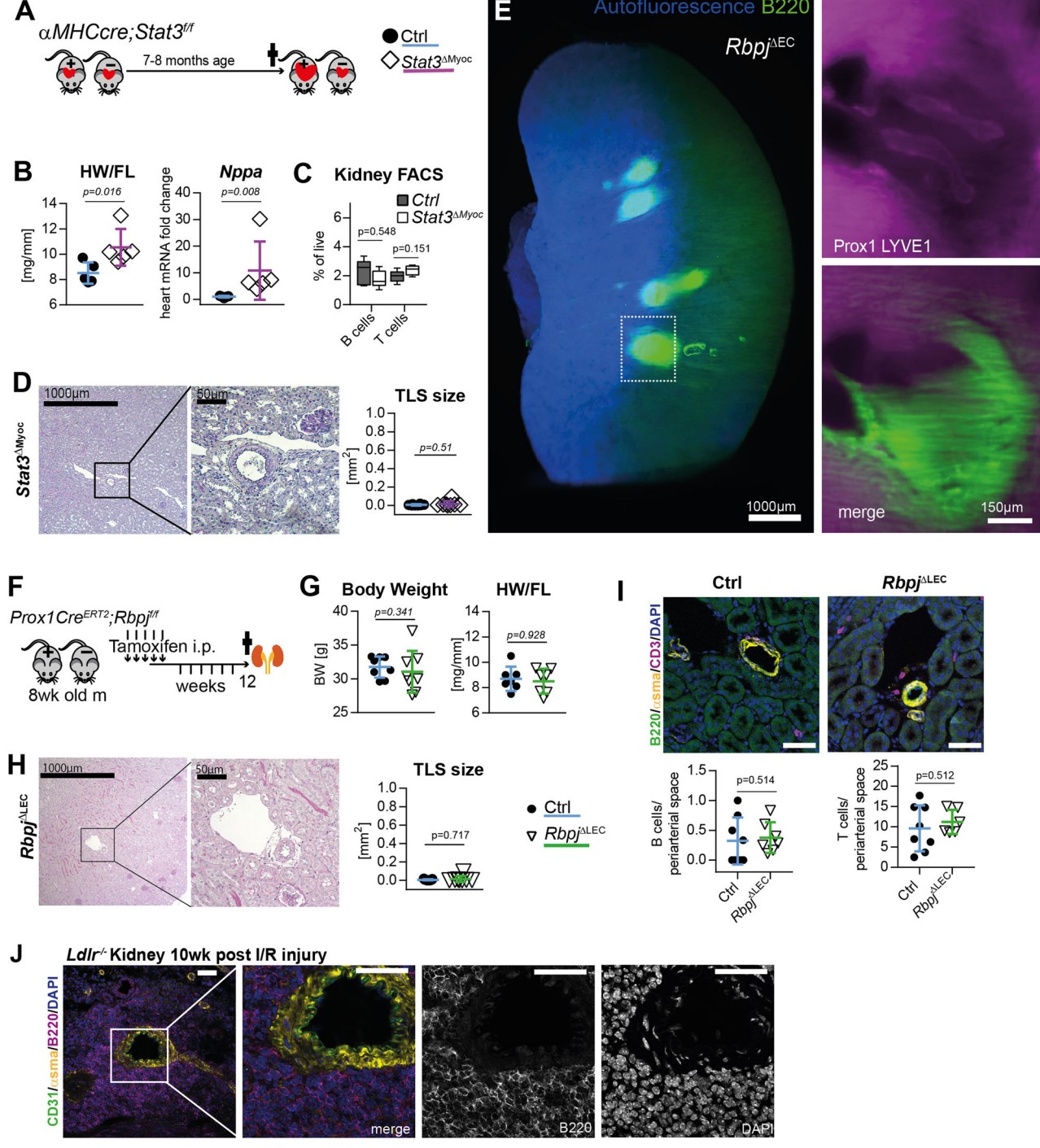

Functionally, HEC create "pockets" for mononuclear cells on their basal side before transmigration[47,48]. Consistent with transformation to HEC functionality, we observed B-lymphocyte clusters underneath the endothelium and within the vascular wall of central arteries of TLS in $Rbpj^{\Delta EC}$ mice (Fig. 4C). Furthermore, whole kidney gene expression analysis revealed robust upregulation of cell-adhesion molecule Madcam-1, another HEV marker, and borderline upregulation of P-selectin (Selp), L-selectin (Sell), and Vcam1 in $Rbpj^{\Delta EC}$ kidneys (Fig. 4D).

To characterize the global gene expression changes of endothelial Notch loss of function we isolated kidney ECs from 3 biological replicates per group (Supplementary Fig. 5A–C) and performed bulk-EC-RNA deep sequencing with >60 million reads

per sample. Principal component analysis revealed predominant clustering by genotypes, corroborating genetic interference in mutant kidney ECs (Fig. 5A). *Hey1* and *Jag1* were downregulated in $Rbpj^{\Delta EC}$ endothelial cells, corresponding to whole kidney QRT results (DeSeq2 results are provided in Supplementary Data file 1, a heatmap of differentially regulated genes in Suppl. data file 4; a Volcano plot with differentially regulated genes in Supplementary Fig. 5D). Using gene set enrichment analysis (GSEA), we found that loss of *Rbpj* in kidney ECs resulted in 1837 gene sets being significantly downregulated, while 123 gene sets were significantly upregulated. A selection of up- and downregulated Gene Ontology (GO) terms with their respective normalized enrichment score (NES) is shown in Fig. 5B (full results in

**Fig. 3 TLS formation in models of heart failure, conditional lymphatic-EC deletion of Rbpj or kidney ischemia reperfusion. A** Experimental set up for analysis of cardiomyocyte restricted deletion of Stat3 (Stat3$^{\Delta Myoc}$). **B** Heart weight to femur length ratio (HW/FL, $p = 0.0159$) and cardiac ANP (Nppa) mRNA expression, $p = 0.0079$; CTRL $n = 5$, KO $n = 5$. Mann–Whitney test, 2-tailed, Graphs: Scatter dot blot, Mean, SD (whiskers). **C** Quantification by lymphocytes by flow cytometry of kidney homogenates; % of live cells, box plots with mean, IQR (25–75%, bounds of box) and total range (min-max: whiskers); CTRL $n = 5$, KO $n = 5$. **D** PAS staining of representative, paraffin-embedded kidney sections. Quantification of infiltrated area [in mm$^2$] per transversal kidney cross-section (sum of all infiltrated areas per section), $N = 8$ biological replicates per group, Mann–Whitney test, two-tailed, $p = 0.51$. Graph: Scatter dot blot, Mean, SD (whiskers). **E** Rbpj$^{\Delta EC}$ Whole kidney staining for B220 (TLS) and Prox1/Lyve1 for lymphatic collecting vessels, light sheet microscopy, ventral view, 3D reconstruction via IMARIS software; scale bar: left image 1000 µm; insets are magnifications of boxed detail, scale bar: 150 µm. Exemplary image, kidneys from $N = 3$ mice stained. **F** Induction protocol for lymphatic endothelial-restricted deletion of Rbpj (Rbpj$^{\Delta LEC}$). **G** Body weight ($n = 8$/group, $p = 0.3409$) and heart weight to femur length ratio ($n = 7$/group, $p = 0.9272$) in 20–22-week-old mice (12 week after KO induction, from two independent experiments); Graphs: Scatter dot blot, mean (box), SD (whiskers). Mann–Whitney test, two-tailed. **H** Paraffin-embedded kidney sections, PAS staining, optical magnification: ×50 left, ×200 detail, scale bar: 1000 µm and 50 µm, as indicated. Quantification of infiltrated area [in mm$^2$] per transversal kidney cross-section (sum of all infiltrated areas per section), $N = 11$ CTRL, $N = 8$ Rbpj$^{\Delta LEC}$ mice per group, Mann–Whitney test, two-tailed, $p = 0.7168$. Graph: Scatter dot blot, mean (box), SD (whiskers). **I** Upper panels: immunofluorescence staining, optical magnification: 200x, scale bar: 50 µm. Lower panels: quantification of periarterial B and T cells per microscopic image (each value = mean #cells per periarterial area of all such areas per one cross-section). CTRL $n = 8$, KO $n = 7$ mice analyzed from 2 independent experiments, Mann–Whitney test, 2-tailed. Graphs: Scatter dot blot, mean (box), SD (whiskers). **J** Immunofluorescence staining as indicated and confocal laser scanning microscopy of kidney 11 weeks after ischemia reperfusion (I/R) injury, optical magnification ×200, scale bar: 50 µm; see also Supplementary Fig. 4; representative picture; $n = 3$ animals in I/R-injury group. Source data are provided as a Source Data file.

Supplementary Data file 2). On whole gene expression level, endothelial cell differentiation gene signatures, matrix organization and EC barrier function along with Notch signaling signatures were significantly downregulated in mutant EC.

To compare the observed gene expression pattern changes of renal EC to previously described gene signatures derived from single-cell RNAseq analysis of various kidney EC populations, we first performed GSEA with defined renal arterial cell transcriptomic signatures[49,50]. In each arterial transcriptomic signature - i. e. large artery, cortical artery, cortical arteriole, medullary arteriole, arteriole efferent—significant and consistent downregulation was registered in Rbpj$^{\Delta EC}$ EC (Fig. 5C). In contrast, GSEA using as gene signature the transcriptomic profiles of homeostatic HEC[12] revealed significant upregulation in mutant EC (NES = 2.3, Fig. 5D). Together, these findings demonstrate that loss of renal endothelial Notch signaling induces an EC phenotype shift characterized by renal arterial dedifferentiation and HEC gene upregulation.

Since our data suggested a close link between loss of endothelial Notch signaling and TLS formation we next analyzed the expression of Notch target genes in various kidney diseases associated with TLS formation in the European Renal cDNA Bank (ERCB) cohort repository[51]. To this end, we compared human kidney biopsy RNAseq datasets from healthy living kidney donors with various forms of inflammatory glomerulonephritis causing either nephrotic syndrome (focal segmental glomerulosclerosis (FSGS), minimal change disease (MCD), membranous glomerulonephritis) or nephritic syndrome (IgA-nephritis, systemic lupus erythematodes (SLE), rapid-progressive glomerulonephritis (RPGN)). Expression of the Notch target gene HES1 was significantly and uniformly decreased in all disease entities compared to living donor (Fig. 5E). Furthermore, while expression of the Notch target HEY1 was significantly downregulated only in FSGS, the Notch target and related gene product HEYL was significantly decreased in all inflammatory kidney diseases except membranous GN. This demonstrates that Notch downregulation is associated with TLS-forming chronic inflammatory kidney diseases, which may indicate involvement in human TLS formation.

Finally, to test whether loss of vascular-arterial EC Notch signaling is sufficient to induce TLS formation, we generated arterial EC-specific BMXCre$^{ERT2}$;Rbpj$^{fl/fl}$ mice (Rbpj$^{\Delta aEC}$)[52] and repeated the experiment (Supplementary Fig. 6A). By flow cytometry we found no increase in CD19+B220+ B cells in

kidney and liver in Rbpj$^{\Delta aEC}$ mice (Supplementary Fig. 6B), nor an increase in dendritic cells, as would be expected in TLS (Supplementary Fig. 6C). Histologically, there were no observable perivascular B cell infiltrates in Rbpj$^{\Delta aEC}$ kidney, liver or heart (Supplementary Fig. 6D). To evaluate Cre-dependent targeting in this adult mouse model we analyzed Cre-dependent reporter gene expression in the kidney of BmxCre$^{ERT2}$; Gt(ROSA)26Sor$^{tm4(ACTB-tdTomato,-EGFP)Luo}$/J mice (BmxCre$^{ERT2}$;mTmG). While EC recombination in proximal segmental arteries was generally high, recombination in second and third-order renal arteries was incomplete (Supplementary Fig. 6E). Thus, our data suggest that loss of proximal arterial EC Notch signaling is not sufficient to induce TLS formation. The role of lower segmental arterial EC Notch signaling in TLS formation deserves further study.

We here show that loss of endothelial Notch signaling in adult mice induces the spontaneous formation of bona fide TLS, based on molecular, cellular, and structural criteria, in several parenchymatous organs. This effect is mediated by the endothelium of blood vessels, but not lymphatics, since a lymphatic-EC-specific targeting strategy did not result in TLS formation; nor was this secondary to heart failure alone. While our screening analysis of mutant mice did not find evidence for TLS formation in the heart, this certainly does not rule out the possibility of TLS formation in the heart, which might occur in structures not included in our analysis or in a different time frame.

TLS formation in mutant mice occurred in a stereotypical manner around second and third-degree renal arteries, which was also observed in several unrelated conditions of kidney injury and during physiological aging in both human and mouse kidneys[10]. This is a so far underappreciated pattern of renal TLS organization. Interestingly, TLS generation has also been described around major non-renal arteries during atherosclerosis development[23,53], which suggests a general mechanism of TLS formation guided by arterial structures. Systematic studies are required to investigate this hypothesis further.

A key role for arteries in the generation of local TLS is also suggested by the fact that arteries in the center of TLS displayed key features of a HEC phenotype: EC decorated with PNAd at the luminal side, leukocyte pockets, integration into FRC conduit networks expressing CXCL13 and, by experimental design, low Notch signature[9,11]. Why cuboidal EC shape was not observed is unclear, but EC height is sensitive to vasoconstriction, pressure, and most importantly, tissue fixation, which could account for the flat appearance of EC[54]. However, apical PNAd expression,

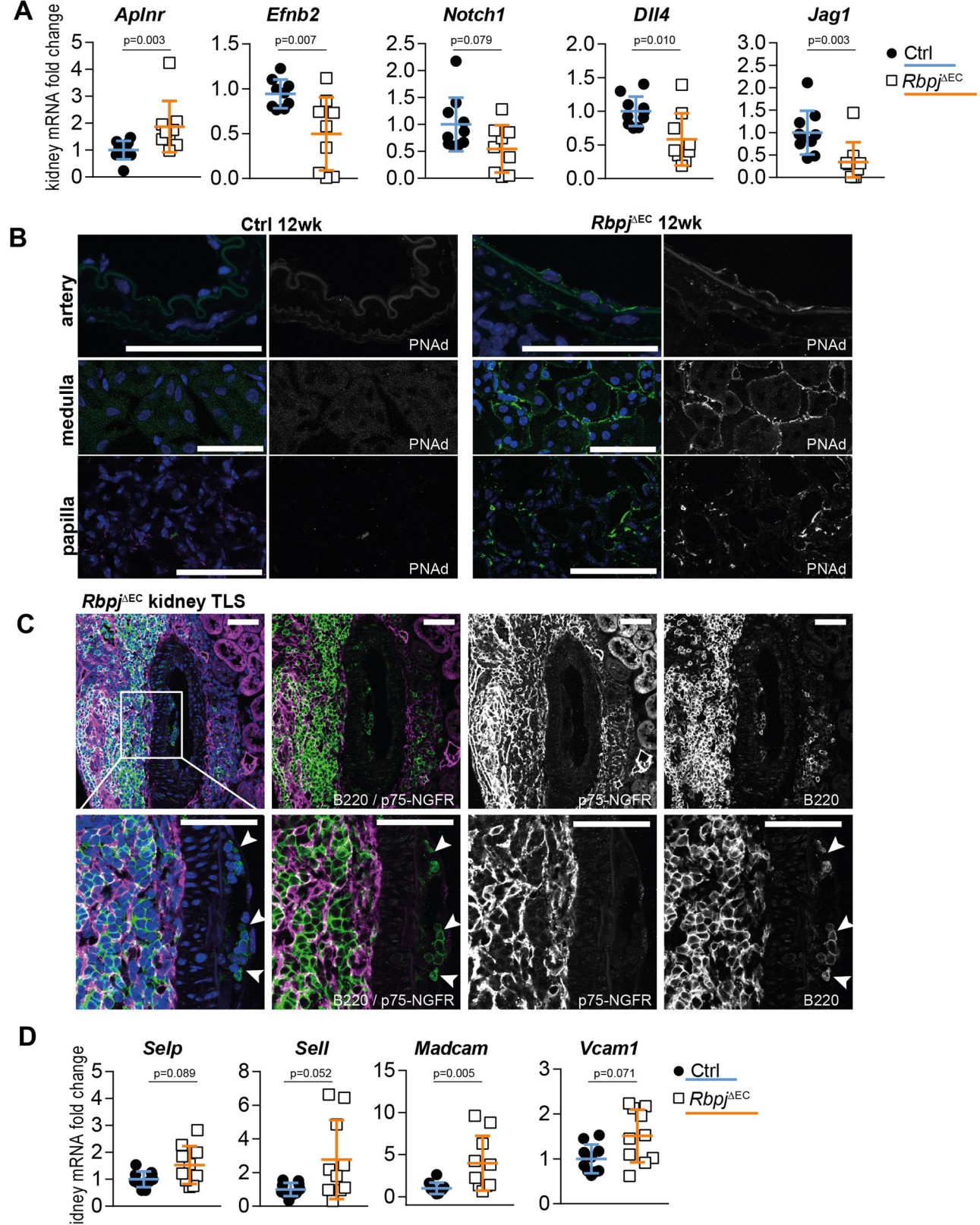

together with the absence of MAdCAM-1, is a key feature of mature HEC in peripheral LN HEV mediating lymphocyte recruitment in a CD62L-dependent manner[55], which, in combination with leukocytes in subendothelial pockets, suggests acquisition of HEC function after Notch loss-of-function. In addition, upregulation of several cell-adhesion molecules, such as

*Sell, Selp, Madcam1,* or *Vcam1*, was observed in whole kidney RNA, which was not reflected in EC transcriptomic analysis (Supplementary Table 1), suggesting that upregulation of these cell-adhesion molecules occurs in recruited leukocytes or a distinct population of reticular stromal cells of lymphoid tissues, which support lymphocyte recruitment[56].

**Fig. 4 Endothelial signatures in TLS development. A** Whole kidney mRNA expression, $n = 10$ mice per group from 2 independent experiments, Mann–Whitney test, two-tailed. Exact p-values: *Aplnr*, p = 0.003; *Efnb2*, p = 0.0068; *Notch1*, p = 0.0789; *Dll4*, p = 0.0101; *Jag1*, p = 0.0033; Graphs: scatter dot blot, mean, SD (whiskers). **B** Immunofluorescence staining and confocal microscopy of *Rbpj^ΔEC* kidney sections. Segmental artery (upper panel), peritubular capillaries (middle panel) and papillary region (lower panel), optical magnification: ×200, scale bar: 50 μm. Representative image, $n = 3$/group stained. **C** Immunofluorescence staining and confocal microscopy of central TLS artery (segmental), inset with higher magnification. B lymphocytes in subendothelial pockets (white arrowheads). Optical magnification: ×200, all scale bars 50 μm. Representative image, $n = 5$/group stained. **D** Whole kidney mRNA expression, relative fold change, $n = 10$/group. Mann–Whitney test, two-tailed. Exact p-values: *Selp* p = 0.089; *Sell* p = 0.052; *Madcam* p = 0.0052; *Vcam* p = 0.071; Graphs: scatter dot blot, mean, SD (whiskers). Source data are provided as a Source Data file.

HEC in secondary lymphatic structures develop after birth and express *Nr2f2*, which encodes the master venous regulator NR2F2 (COUP-TFII), a repressor of Notch signaling[19]. At the same time, genes associated with arterial specification, e. g. *Efnb2, Notch1, Dll4,* and several Notch target genes, are uniformly downregulated in HEC, when compared to capillary EC[11]. Conversely, active Notch signaling induces and maintains, in a dose-dependent manner, an arterial EC phenotype[15,44,45,57,58]. In fact, Notch signaling activity defines a developmental trajectory from venous to capillary to early/late arterial EC, in which HECs align to the low Notch signaling spectrum[11,45].

Maintenance of arterial phenotype requires continuous Notch signaling activated by arterial shear stress, which mediates anti-inflammatory effects of Notch signaling, while deletion of Notch1 induces loss of arterial specification and disruption of anti-inflammatory gene networks, which promotes pro-inflammatory responses of EC and atherosclerosis development[44,57,59,60]. Our finding of a loss of an arterial gene signature in the transcriptome of *Rbpj^ΔEC* mice—specifically shown in kidney endothelium via RNAseq and GSEA—corroborates the notion of active maintenance of arterial phenotype by Notch signaling, but also demonstrates an EC phenotype shift towards HEC with development of TLS in the kidney. This also provides another basis for the pro-inflammatory effects and atherosclerosis development observed in mice with disrupted endothelial Notch signaling, since atherosclerosis is accompanied by arterial TLS formation[23,57,60]. Furthermore, the reciprocal link between Notch signaling and vascular inflammation[61,62] might provide a conceptual framework to understand TLS formation via an EC phenotype shift. Pro-inflammatory cytokines and lipids downregulate Notch signaling in arterial EC in vitro and in vivo[60], which in responsive EC might induce a HEC phenotype shift and local development of TLS. Our data therefore support the hypothesis of an antigen-independent mechanisms of TLS generation mediated by prolonged inflammation[2], which, via downmodulation of Notch signaling and HEC induction, promotes TLS formation. At the same time, this also implies a novel aspect of endothelial dysfunction, a key driver of vascular disease[63]. A molecular link between loss of Notch signaling, inflammation, and upregulation of PNAd expression is provided by the interaction with NF-kB, which is suppressed by Notch signaling and required for proper PNAd expression and HEC phenotype[64,65].

Our analysis also revealed that secondary lymphoid tissues, such as lymph nodes and spleen, were largely unaffected in mutant mice, judged by size and cell numbers. A small but significant increase in dendritic cell numbers was noted, which is consistent with TLS development, due to their role in TLS formation[66]. Disruption of endothelial Notch signaling is expected to have the greatest impact in vascular beds with high Notch signaling activity, e. g. arteries and certain capillary beds, but little effects in veins or HEV with low Notch signaling activity[11,45]. Our data therefore are in line with the hypothesis that change in the arterial signature is involved in TLS formation. The capacity to form TLS represents a sustained form of an adaptive immune response arising de novo in affected organs[67–69]. The functional dependence on arterial signature changes, which involves downmodulation of Notch signaling by inflammatory conditions and regional conversion to HEC phenotype, could represent an evolutionary conserved mechanism to assemble lymphoid structures in affected organs, since Notch signaling is an ancient and evolutionary conserved signaling pathway. Our observation also provides a rationale for the described association of old age and development of TLS[10], since Notch signaling components and Notch-dependent arteries and vascular networks in the bone decrease with age. This decrease can be rescued by endothelial overexpression of Notch, suggesting impairment of endothelial Notch signaling with age[70]. Lastly, the fact that TLSs develop without alteration of secondary lymphoid structures (in terms of size and cell composition, aside from dendritic cell content) even suggest a potential therapeutic angle specific for TLS without affecting secondary lymphoid structures, and thus general immunosuppression.

## Methods

**Mice**. *Cdh5Cre^ERT2*; *Rbpj^f/f* (B6-Tg(Cdh5(PAC)-cre/ERT2)^1Rha Rbpsuh ^tm3Hon/ Rbrc) were generated by crossing *Cdh5(PAC)-CreERT2* mice[71] with *Rbpj^f/f* mice[20]; *BmxCre^ERT2*;*Rbpj^f/f* mice (B6-Tg(Bmx(PAC)-cre/ERT2)^1Rha Rbpsuh ^tm3Hon/Rbrc) were generated by crossing *Bmx(PAC)-CreERT2^1Rha*,[52] mice with *Rbpj^f/f* mice[20]. *Cdh5Cre^ERT2*;*TdTomato^+/+* mice were generated by crossing *Cdh5(PAC)-CreERT2* mice[71] with B6.Cg-Gt(ROSA)26Sortm14(CAG-tdTomato)Hze/J (Jackson Laboratories Strain #:007914).

*αMHCCre*;*Stat3^f/f* mice were described before[39]. *LDLr^−/−* mice were purchased from Jackson Laboratories (strain 002207,[72]). All mice were housed under specific pathogen-free conditions (Type 22 polysulfon IVC systems with positive pressure) in the animal facility of Hannover Medical School with a 14/10 h light/dark cycle (generated by a 400 lx light source), $21 \pm 2\,°C$ ambient temperature and $50 \pm 5\%$ relative humidity, and were supplied with autoclaved water and food (Altromin TPF-1324) ad libitum. All experiments were performed with male mice and age-matched littermate controls with the approval of the local animal welfare board (LAVES, Lower Saxony, Animal Studies Committee, protocol numbers 15/1944, 18/2931, and 18/2973; breeding and echocardiography on *αMHCCre*;*Stat3^f/f* on protocols 33.12-42502-04-18/2807 and 33.19-42502-05-18A271).

*BMXCre^ERT2*;*mTmG* mice were generated by crossing *BMX(PAC)-Cre/ ERT2^1Rha*,[52] mice with *Gt(ROSA)26SOR^tm4(ACTB-tdTomato-EGFP)Luo/J* mice[73]. A recombination control experiment was performed with approval of LANUV, Northrhine-Westphalia, protocol number Az 81-02.04.2019.A114.

*Prox1Cre^ERT2*; *Rbpj^f/f* mice were generated by crossing Prox1-ERT2 mice[43] with *Rbpj^f/f* mice[20]; and *Prox1Cre^ERT2*;*mTmG* mice were generated by crossing Prox1CreERT2 mice[43] with *Gt(ROSA)26Sor^tm4(ACTB-tdTomato,-EGFP)Luo/J* mice[73]. Mice were provided water and food (Scientific Animal Food & Engineering, R150) ad libitum. Mice were on a 12 h light/dark cycle and kept at $22\,°C \pm 2\,°C$ with a relative humidity of $55\%\pm10\%$. Experiments with these mice were performed with the approval of the Animal Ethics Committee of Vaud, Switzerland, protocol number VD2914.

**Experimental procedures**. At 7–9 weeks of age, *Cdh5Cre^ERT2*; *Rbpj^f/f* mice or Cre-negative littermate controls were injected 1500 μg Tamoxifen i.p. (dissolved in medium chain fatty acids at a dose of 10 mg/ml, volume 150 μl/dose) on five consecutive days and euthanized at the indicated time points. *Prox1Cre^ERT2*; *Rbpj^f/f* or Cre-negative littermate controls were injected 1500 μg Tamoxifen i.p. (dissolved in medium chain fatty acids at a dose of 10 mg/ml, volume 150 μl/dose) on five consecutive days. Kidneys were collected and fixed in 4% PFA for histological analysis and immunostaining.

*αMHCCre*;*Stat3^f/f* mice or Cre-negative littermate controls were euthanized at 7–8 months of age after verification of heart failure phenotype via echocardiography in sedated mice (2% isoflurane inhalation, connected to a rodent ventilator) using a

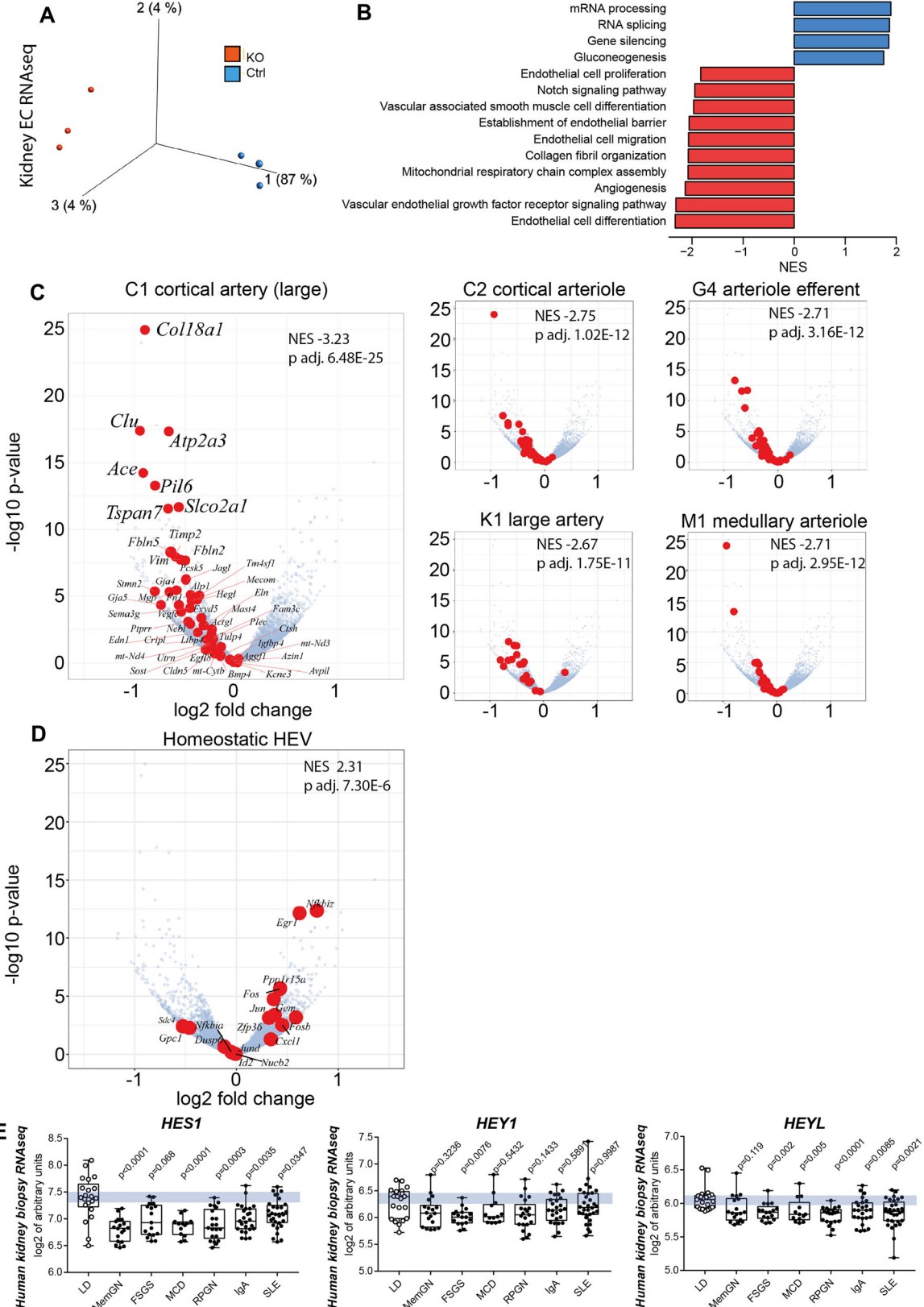

Vevo 770 (Visual Sonics) as described[74] and tissues were collected for further analysis. Male LDLr⁻/⁻ mice underwent unilateral ischemia reperfusion injury (27 min ischemia) of the kidney at 8-10 weeks of age[75] and were fed a high fat diet (C1090-70) for 10 weeks starting 1 week after surgery.

All animal studies were undertaken in accordance with German Animal Welfare legislation and with the European Communities Council Directive 2010/63/EU for the protection of animals used for experimental purposes. All

experiments were approved by the Local Institutional Animal Care and Research Advisory Committee and permitted by the relevant local authority for animal protection.

**Tissue and cell preparation for flow cytometry**. Mice were sacrificed and blood was collected from the Vena cava in Na2EDTA containing tubes; spleens, kidneys,

**Fig. 5 Kidney endothelial cell RNAseq and gene set enrichment analysis (GSEA). A** Principal Component Analysis of $Rbpj^{\Delta EC}$ vs. Control kidney EC transcription analysis, $n = 3$/group, (Variance filtering 0.05, Student's T test followed by the B-H correction ($p < 0.02$, FDR $< 0.0199731$) showing biological replicates of each group clustering together (370 genes) (more information in "Methods"). **B** GO-term GSEA—selected significantly up- (red) and downregulated (blue) gene sets sorted by normalized enrichment score (NES). **C** Volcano plots of individual genes with expression changed in $Rbpj^{\Delta EC}$ compared to control, with $\log_2$ (fold change) on x-axis and $-\log_{10}$ (adjusted p-value) on the y-axis; in red, genes belonging to selected marker gene sets for different kidney arterial segments (from refs. [49,50]). NES and p value upper right corners; see "Methods" (GSEA) for statistics. **D** Homeostatic HEC marker gene set from[12] in red in same volcano plot, NES and p value upper right corner. See Methods (GSEA) for statistics. **E** Human kidney biopsy RNAseq (GSE: EC Notch target genes in different glomerular diseases downregulated as compared to living kidney donor biopsies as control. Abbreviations and sample number: LD, living kidney donor ($n = 21$); MemGN, membranous glomerulonephritis ($n = 18$); FSGS, focal segmental glomerulosclerosis ($n = 17$); MCD, minimal change disease($n = 13$); RPGN, rapid-progressive glomerulonephritis ($n = 21$); IgA, Iga-Nephritis ($n = 25$); SLE, systemic lupus erythematodes ($n = 32$). Single samples (dot) plus mean, IQR (box) and total range (min-max: whiskers). Statistic: Brown-Forsythe and Welch (1 way) ANOVA with Dunnet's multiple comparisons; exact adjusted p-values: *HES1*, compared to LD: MemGN $<0.0001$; FSGS, 0.068; MCD, $<0.0001$; RPGN, 0.0003; IgA, 0.0035; SLE, 0.0347. *HEY1*, compared to LD: MemGN, 0.3236; FSGS, 0.0076; MCD: 0.5432; RPGN, 0.1433; IgA, 0.5891; SLE, 0.9987. *HEYL*, compared to LD: MemGN, 0.119; FSGS, 0.002; MCD, 0.0050; RPGN, $< 0.0001$; IgA, 0.0085; SLE, 0.0021. Source data are provided as supplementary files.

---

bones and para-aortic lymph nodes were excised and kept on ice during preparation. Spleens and lymph nodes were pressed, resuspended in PBS (Millipore) and filtered through a 70 μm mesh; blood and bone marrow from tibia and femur were filtered as above. Erythrocytes were removed from splenic and blood cell suspensions by red blood cell lysis buffer (Biolegend).

Kidneys were minced to small pieces (<1 mm), then digested with Collagenase II (Worthington) 500U/ml for 2× 21 min at 37 °C, interrupted by 1–2 courses of tissue dissociation (GentleMACS, program B). Cells were washed with PBS and filtered through a 70μm mesh several times before proceeding with staining. For CD23 staining experiments, kidneys were pressed and filtered as indicated above and lymphocytes were isolated using double (70–40%) percoll (GE Healthcare) density gradient centrifugation. After extensive washing cells from all organs were resuspended into PBS, counted using a Countess II automatic cell counter (Thermofisher Scientifics) and used for flow cytometry.

**Flow cytometry**. Cells were resuspended in buffer containing 2% fetal calf serum (Biochrom), 2 mM Na2EDTA (Roth) and 0.05%NaN3 (AppliChem) and stained using antibodies listed in the table after Fc receptor blocking with TrueStain fcX anti-CD32/16 (Clone 93, Biolegend). Biotinylated antibodies were bound by Streptavidin PE-Dazzle 594.

Propidium Iodide (Fluka) was used to exclude dead cells. Cells were analyzed on LSR II flow cytometer (BD Biosciences, acquisition software BD FACSDiva Software v8.0.1) and data were analyzed using FlowJo software v8.0.1 (TreeStar). List of antibodies and dilutions used for flow cytometry in Supplementary Table 1.

**Kidney endothelial cell isolation**. For purification of EC from kidneys, we combined MACS-based pre-enrichment and FACS strategies. Single-cell suspensions were prepared from kidneys as described above. After Fc block and subsequent staining with anti-CD31-PE and anti-CD45-FITC, cells were washed and incubated with 1:5 diluted anti-PE microbeads (Miltenyi Biotech, 130-048-801) for 15 min. Positive selection of magnetically labeled cells was performed using LS columns and MidiMax Separator in combination with MACS Multistand (all from Miltenyi Biotec) according to manufacturer's instructions. After elution from columns, cells were pelleted, filtered through 40 μm mesh and proceeded for sorting on a FACSAria IIu cell sorter (Becton-Dickinson). Total yield of EC (CD31+CD45− cells) was about $5 \times 10^5$ to $1 \times 10^6$ live cells per kidney. RNA was isolated directly after sorting using a Qiagen RNEasy micro kit according to the manufacturer's protocol.

**RNA sequencing**

*Library generation*. Two nanograms of total RNA were used for library preparation with the 'SMARTer Stranded Total RNA-Seq Kit v2 – Pico Input Mammalian' (#634413; Takara/Clontech) according to conditions recommended in user manual #063017. Generated libraries were barcoded by dual indexing approach and were finally amplified with 12 cycles of PCR. Fragment length distribution of generated libraries was monitored using the 'Bioanalyzer High Sensitivity DNA Assay' (5067-4626; Agilent Technologies). Quantification of libraries was performed by use of the 'Qubit® dsDNA HS Assay Kit' (Q32854; ThermoFisher Scientific).

*Sequencing run*. Equal molar amounts of six libraries in total were pooled for a common sequencing run. Accordingly, each analyzed library constitutes 16.6% of overall flowcell capacity. The library pool was denatured with NaOH and was finally diluted to 2 pM according to the Denature and Dilute Libraries Guide (Document # 15048776 v02; Illumina). A volume of 1.3 ml of denatured pool was loaded on an Illumina NextSeq 550 sequencer using a High Output Flowcell for single reads (20024906; Illumina). Sequencing was performed with the following settings: Sequence reads 1 and 2 with 38 bases each; Index reads 1 and 2 with 8 bases each.

*BCL to FASTQ conversion*. BCL files were converted to FASTQ files using bcl2fastq Conversion Software version v2.20.0.422 (Illumina).

*Raw data processing and quality control*. Raw data processing was conducted by use of nfcore/rnaseq (version 1.4.2) which is a bioinformatics best-practice analysis pipeline used for RNA sequencing data at the National Genomics Infrastructure at SciLifeLab Stockholm, Sweden. The pipeline uses Nextflow, a bioinformatics workflow tool. It pre-processes raw data from FastQ inputs, aligns the reads and performs extensive quality control on the results. The genome reference and annotation data were taken from GENCODE.org (Mus musculus; GRCm38.p6; release M25).

*Normalization and differential expression analysis*. Normalization and differential expression analysis were performed with DESeq2[76] (Galaxy Tool Version 2.11.40.2) with default settings except for "Output normalized counts table", "Turn off outliers filtering", and "Turn off independent filtering", all of which were set to "True".

Output counts were used for further analysis with Qlucore Omics explorer (Sweden). Data were log2 transformed, 1.1 was used as a threshold and low expression genes (<30 reads in all samples) were removed from the analysis.

Principal component analysis (PCA) was performed on 370 differentially expressed genes (DEGs) after variance filtering (filtering threshold 0.05) selected by two-tailed Student's t test with the Benjamini-Hochberg (B-H) correction ($p < 0.02$, FDR $< 0.0199731$). The DEG list was used for generation of PCA plot using built in functionality in Qlucore omics explorer with default settings.

**Gene set enrichment analysis (GSEA)**. GSEA was conducted according to the method described in 2005[77] using the gene sets of the Gene Ontology (GO) Biological Processes resource[78,79]. First, genes were filtered to only retain the ones that had a mean expression level above $\log_2(\text{normalized counts}+1) = 3.23$, which yielded 18,862 genes. Next, the ordered Wald statistics calculated by DESeq2 were provided to the "gseGO" function of the clusterProfiler package (v. 4.0.5[80]) in R (v. 4.0.1), using parameters eps = 1e-60, minGSSize = 30, and maxGSSize = 2000. With this method, an enrichment score (ES) was calculated for each gene set by decreasing or increasing a Kolmogorov-Smirnov statistic according to the magnitude of the Wald statistic of each gene (using $p = 1$ as in Equation 1 of Subramanian et al, 2005). One thousand permutations were performed to obtain randomized ES and calculate the normalized enrichment score (NES) by dividing the real ES by the mean of the randomized ES values. The p-value associated with each gene set was adjusted by using the Benjamini-Hochberg procedure[81].

The list of significant gene sets with adjusted p-value < 0.05 was manually parsed and representative GO terms were selected to create a barplot of normalized enrichment scores (full GSEA results in Supplementary Table 1). Finally, we used the same methods and functions to calculate enrichment scores, normalized enrichment scores and p-values for a custom list of gene signatures that we compiled from kidney endothelial single-cell RNAseq publications, defining arterial EC or HEC[12,49,50].

**Gene array data analysis**. Tubulointerstitial *Hey1*, *Hes1*, and *HeyL* expression were analyzed in public datasets from the European cDNA bank cohort, the Nephrotic Syndrome Study Network, and the Vasculitis Clinical Research Consortium[51] obtained at NCBI GSE104948, and GSE104954.

**Quantitative real-time PCR analysis**. Total RNA was purified from cell lysates using Nucleospin RNA II kit (Macherey Nagel). After purity and quality check, RNA was transcribed into cDNA employing a cDNA synthesis kit (Invitrogen) according to the manufacturer's instructions. Quantitative real-time PCR was performed using specific primers (see Supplementary Table 5) with FastStart

Essential DNA Green Master Mix on a LightCycler 96 system from Roche (acquisition software: LightCycler96 Version 1.1.0.1320 (© 2011 Roche Diagnostics International)) according to the manufacturer's instructions. The expression of each specific gene was normalized to the expression of *Rps9* and calculated by the comparative CT ($2^{-\Delta\Delta CT}$) method[82]. Primer sequences in Supplementary Table 2.

**Tissue fixation, embedding, and stainings**. Immunohistochemistry and immunofluorescence staining in mice were performed as previously described[18] with modifications. Mice were euthanized; kidney, liver, lung, spleen, heart and mesenteric lymph nodes were isolated and fixed in 4% buffered paraformaldehyde (PFA). Organs were either embedded in paraffin or cryopreserved in sucrose and embedded in Tissue-tek O.C.T. compound (Sakura, Germany). Paraffin blocks were cut with a rotation microtome (Leica) at 1–2 μm thickness and stained according to routine histological protocols (hematoxylin eosin, periodic acid Schiff, Masson trichrome, Sirius red).

O.C.T.-embedded kidneys were cryosectioned into 6-μm sections and mounted on Superfrost slides (Fisher Scientific). For immunofluorescence and laser scanning microscopy, sections were washed in 1X PBS, blocked in 10% normal donkey serum (Vector Laboratories), and incubated with primary Abs (against PDPN, Clone RTD4E10, abcam ab11936, dilution 1:400; Lyve1, R&D, BAF2125, dilution 1:100; B220, Clone RA3-6B2, Life Technologies 14-0452-82, dilution 1:100; CD3, DAKO A0452, dilution 1:100; CXCL13, R&D, AF470, dilution 1:1000; KI67, Thermo Fisher 14-5698-82, dilution 1:100; NG2, Millipore #AB5320, dilution 1:100; ER-TR7, BMA, dilution 1:100; Rankl, Clone IK22/5, eBioscience 14-5952-81, dilution 1:100; PNAd, Clone MECA79, Biolegend, dilution 1:100; Erg, Clone EPR3864, Abcam ab92513, dilution 1:400, Prox1, R&D, AF2727, dilution 1:100; CD31, Clone 390, Biolegend 122407, dilution 1:100; anti-GFP, AvesGFP-1010, dilution 1:500) and appropriate fluorescence labeled secondary antibodies (Anti-host, AF488, AF555 or AF647, Life Technologies, dilution 1:500) or with directly labeled antibodies (Anti-asma-AF488, Clone 1A4, BD biosciences, dilution 1:300; asma-Cy3, Clone 1A4, Sigma C6198, dilution 1:300; GL7-FITC, Biolegend 144603, dilution 1:100; IgD-AF647, Clone 11-26c.2a, Biolegend #405707, dilution 1:100; CD21/35-FITC, Clone 7G6, BD Pharmingen 561769, dilution 1:100). DAPI (4,6-Diamidino-2-phenylindole, Invitrogen, Germany; dilution 1:2000) was used for counterstaining of nuclei and slides were mounted in Immunoselect Antifading Mounting Medium (Dianova, Germany). For CXCL13 staining, a TSA-Cy3 signal amplification kit was used (SAT704A001KT, Perkin Elmer, USA). Images were acquired using Leica TCS SP8 AOBS (Leica Microsystems, Germany) confocal microscope with a ×20 objective or Zeiss LSM980 confocal microscope (Zeiss, Germany) with a ×25 objective.

**Quantification of TLS area**. Kidneys were cut in half along the axial plane and mounted face down, resulting in sections representing the middle of the kidney with hilar, papillar, medullary, and cortical fractions. In PAS stained sections, area covered by mononuclear cells was measured in the periarterial loose interstitial tissue area (vessel nerve sheath) along all (typically longitudinally cut) segmental and all (typically cross sectioned) interlobar arteries per section. As the kidneys/ sections were of similar size, we did not normalize the measured area to the section size.

**Whole-mount staining and optical clearing**. For CD31-staining, we directly labeled anti-CD31-antibody (clone 390, Biolegend) with fluorochrome AF790 using a direct labeling kit from Thermo Fisher Scientific (Cat: A20189). Five micrograms of antibody in PBS were injected via the tail vein. After 10–30 min, mice were euthanized, perfused through the left ventricle with PBS + 2 mM EDTA and then with buffered 4% PFA.

For whole-mount staining, we followed an adapted iDisco-Protocol[83] with an added bleaching step and cleared the organs with ethyl cinnamate (ECI[84]). In short, kidneys were harvested and postfixed in 4%PFA for 2 h; dehydrated in ethanol (50%, 70%, 100%) bleached with ETOH with 5% DMSO/ 5%H2O2; rehydrated stepwise; washed; incubated with permeabilization solution[83] at 37 °C for 2 days; blocked with blocking solution[83] at 37 °C for 2 days; Incubated with 5 μg/ml anti-B220-AF647 (Clone RA3-6B2, Biolegend #103226) in PTwH/ 5%DMSO/ 3% Donkey Serum, 37 °C, 3 days. Washed 4–5× until the next day; dehydrated in ethanol/H2O series and then cleared in ECI at room temperature. For the B220 and Prox1/LYVE1 costaining, we incubated with anti-Prox1 (R&D AF2727, 1:100) and Anti-LYVE1 (R&D BAF2125, 1:100) for 2 days, washed for 24 h, and incubated with anti-goat-Cy3 (1:500) and anti-B220-AF647 (1:100) for another 2 days at 37 °C. Imaging using light sheet microscopy; for CD31/B220, we used a LaVision BioTec Ultramicroscope (Imaging Center Essen), for B220/Prox1, Lyve1 a LaVision BioTec UltraMicroscope II at Hannover Medical School (different wavelength laser equipment). For 3D Image reconstruction and movie animation, Imaris Software 7.6.5 Version I (Bitplane/Oxford Instruments) was used.

Serum and urine measurements were performed with an automated Olympus AU400.

**Statistics**. Results are expressed as box plots with min/max whiskers, mean and 25–75%-range (box) or as scatter dot plots with mean ± standard deviation. *N* numbers are biological replicates of experiments performed at least three times

unless otherwise indicated. The significance of differences was calculated using the tests indicated in the respective figure legend (two-tailed Mann–Whitney test unless otherwise stated) with confidence interval of 95%.

**Reporting summary**. Further information on research design is available in the Nature Research Reporting Summary linked to this article.

## Data availability
The RNA sequencing data generated in this study, including raw sequencing files and a table of preprocessed counts per gene per sample, are publicly available in the NCBI's Gene Expression Omnibus under accession number GSE193544. The mouse reference genome and annotation (GRCm38) used to align sequencing reads are available from GENCODE (https://www.gencodegenes.org/mouse/release_M25.html), and the Gene Ontology gene sets are available here: http://geneontology.org/.

Tubulointerstitial Hey1, Hes1, and HeyL expression were analyzed in public datasets from the European cDNA bank cohort, the Nephrotic Syndrome Study Network, and the Vasculitis Clinical Research Consortium[51], obtained at NCBI: GSE104948, and GSE104954.

All other data are provided in the Supplemental Information, Supplemental Data files and the Source Data file accompanying this article. Source data are provided with this paper.

## Code availability
All bioinformatic tools and methods used in this manuscript have been published previously; we therefore did not deposit any code in a public repository. For further information please contact the corresponding author. Source data are provided with this paper.

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

## Acknowledgements

Schematics were created with BioRender.com. We thank the Central Animal Facility, Research Core Facility Cell Sorting and Research Core Unit Laser Microscopy of Hannover Medical School for support. We thank Stefan Sablotny, Michaela Beese, Anja Standke, Herle Chlebusch, Birgit Brandt, Anja Bubke, and Petra Berkefeld for excellent technical support. We thank Zulrahman Erlangga for his help optimizing our kidney EC isolation protocol, and Heiko Schenk for his help in obtaining a chicken-anti-GFP antibody. We thank Taija Makinen for providing Prox1-CreERT2, Freddy Radtke for *Rbpj*^f/f mice and Ingmar Mederacke for providing B6.Cg-Gt(ROSA)26Sortm14(CAG-tdTomato)Hze/J (Jackson Laboratories Strain #:007914) mice for breeding. Funded by intramural grants from Hannover Medical School (Hochschulinterne Leistungsförderung II, Clinical Scientist Program "Junge Akademie", and Ellen-Schmidt-Habilitationsförderung ESP) to S.F., and by grants from Deutsche Forschungsgemeinschaft to T.K. (Ka5549/2-1), to S.v.V. (VI508/7-1), to J.G. (Ga2443/3-1), and to F.P.L. (Li948-7/1), as well as DFG KFO311 (HI 842/10-2 to D.H.-K. and RI 2531/2-2 to M.R.-H.) for the Stat3^ΔMyoc-experiment. DFG-funded CRC 1348 (Dynamic Cellular Interfaces) to R.H.A. Grants by the Swiss League for Cancer Research (KFS-4895-08-2019, to T.V.P.) and the Swiss National Science Foundation (CRSK-3_190435 to J.B.-L.).

## Author contributions

S.F., T.K., J.B.-L., J.G., J.L., D.K., E.B., A.H., S.H. did the experiments; S.F., T.K., F.P.L. designed experiments and analyzed data; D.H.K., M.R.H., T.P., D.R.E., S.v.V., R.H.A., H.H. provided necessary materials or animals; T.W., J.G., B.L., and A.C.J. provided bioinformatic analyses; S.F. and F.P.L. wrote and edited the manuscript; All authors read, commented and corrected the manuscript; F.P.L. conceived and directed the study.

## Funding

## Competing interests

The authors declare no competing interests.
