## [Peer Review File · Nature Communications]

REVIEWER COMMENTS

Reviewer #1 (Remarks to the Author):

This manuscript addresses an important issue as to the mechanisms underlying the formation of TLS. The authors show that endothelial cell-specific deletion of *rbpj* leads to TLS in murine kidneys. The kidney TLS are located around arteries. The demonstration that the *cdh5* notch signaling pathway is involved in TLS formation is interesting and potentially important.

However, there are several major issues with the body of evidence presented that need to be addressed.

- 1. *Cdh5*-specific deletion of *rbpj* is an EC phenotype most likely in all endothelial cells of all organs including the brain. The authors observe, however, TLS specifically in the kidney: No mechanism is presented or discussed why TLS form in the kidney.**
- 2. A comprehensive TLS search evaluation should be performed in other major organs including the heart, the liver, the lung and the brain and more to show that the observation is kidney-specific.**
- 3. The authors claim that secondary lymphoid organs remain unchanged but they do not provide comprehensive evidence for this claim. What do they mean by "unchanged" ? They do not provide evidence that SLOs are unchanged other than selective (biased) analyses. A claim of this nature should be substantiated by genome wide expression analyses of lymph nodes and spleen preferably using single cell transcriptome analyses with and without *rbpj* deletion in ECs and immune cells.**
- 4. The authors observed 12 weeks after knockout higher frequencies of mature B cells in the kidney and lower frequencies in blood and the bone marrow and attribute these changes to a recruitment phenomenon into the kidney without evidence for this interpretation. There are many other possibilities to interpret these data, i.e. effects of the knockout in the bone marrow itself including its stem cell niche, immune cells and mesenchymal cells. The authors need to rule out these possibilities.**
- 5. The authors show that there is an increase in dendritic cells in the kidney, bone marrow and spleen. These data appear to contradict their claim that the immune system only responds to *Cdh5* *rbpj* signaling deficiency to recruit immune cells into kidney arteries but rather appears to indicate that the knockout is associated with major changes in the immune system itself possibly affecting the bone marrow and SLOs.**
- 6. The authors perform selective transcript analyses in HEC. These analyses are highly biased and are not sufficient to provide comprehensive and sufficient information on the phenotypes of the HEC in response to *rbpj* deletion in endothelial cells. The authors should perform single cell transcript analyses of sorted endothelial cells in major organs including the kidney with and without knockout to get a more complete picture of the phenotype of the various EC subtypes as expressed for example in t-SNE projections.**

Reviewer #2 (Remarks to the Author):

Fleig and colleagues present an interesting set of data supporting a role for endothelial Notch signalling in the formation of tertiary lymphoid structures in the kidney. The authors show that loss of the Notch signalling mediator *Rbpj* in endothelial cells, but not lymphatics, leads to tertiary lymphoid structures around renal arteries. The authors postulate that this phenotype involves a loss of arterial specification and acquisition of a "high endothelial cell" phenotype. The

work presented is novel and an important finding, tertiary lymphoid structures are found in several renal diseases, but the molecular mechanisms are completely unknown. This paper begins to address this issue. Although of interest, there are several areas where the paper could be strengthened, in particular details about the deletion of *Rbpj* in the kidney endothelium, the description of histological changes in the kidney and the strategy used to argue that cardiac problems do not contribute to the renal phenotype as outlined below.

1. The authors take the strategy of deleting *Rbpj* using a *Cdh5* promoter. However, there is considerable heterogeneity in the endothelium of the adult kidney. Do the authors have any insights into the specific vascular beds where *Cdh5* is expressed in the normal adult kidney? Are they targeting large arteries, glomerular capillaries, peritubular capillaries, vasa rectae, lymphatics?
2. Limited data is provided to support the successful deletion of *Rbpj* in the kidney endothelium. Currently, only a qRT-PCR of *Hey1* on the whole kidney is provided. Can information ideally on isolated endothelial cells examining *Rbpj* expression levels be provided? There is also little information in the literature on the expression of *Rbpj* and *Hey1* in adult kidneys. Can the authors provide any information - if *Rbpj* levels are already low in adult kidneys, then what is the rationale for dampening signalling further?
3. The authors present histological observations indicating the presence of tertiary lymphoid structures in the mice with deletion of endothelial *Rbpj*. It would be good to have a bit more information here, how many of these structures are seen in each mouse and where are they localised in the kidney (cortex/medulla, close to glomeruli). The lymphoid structures are described as being around arteries, but this is not clear to the reviewer in some of the pictures presented (e.g. mutant samples in Figure 1C, top panel). How is the size of tertiary lymphatic structures being assessed, what is examined in a control animal?
4. A degree of renal injury is indicated in the endothelial knock-out of *Rbpj*. Again, it would be optimal to see some more histological evidence to support this. Is the structure of the glomerulus altered or are there any signs of tubular injury? How about more specific functional markers such as albuminuria, creatinine clearance or blood urea nitrogen.
5. Is the presence of lymphoid tertiary structures a renal-specific effect or is there any evidence of similar structures in other non-lymphoid organs?
6. The authors argue that cardiac failure may contribute to the development of tertiary lymphoid structures in their mice. However, it looks like cardiac failure occurs later than the time-points where the kidney study has been performed. What do the hearts look like at earlier time-points? Instead, a different strategy examining renal tertiary lymphoid formation in mice lacking *Stat3* in the myocardium who develop heart failure is taken. No lymphoid structure formation in the kidney is found but it is difficult to see how the authors can really relate this different model to the observations they see in mice lacking endothelial *Rbpj*. Does it really rule out cardiac failure driving the changes seen in renal tertiary lymphoid structures in their mice?
7. In Figure 3E, pictures of lymphatic structures are shown. If possible, the findings would be strengthened if the authors could show a movie of their data here.
8. The authors go on to show that renal tertiary lymphoid structures do not form in mice lacking *Rbpj* in *Prox1* expressing cells. The authors should note some caveats here as *Prox1* is not solely expressed in lymphatics in the kidney (Kim YM, *PLoS One*. 2015; 10(5):e0127429; Kenig-Kozlovsky Y et al *J Am Soc Nephrol*. 2018 Apr; 29(4):1097-1107). As highlighted above, data should be provided to show the transgenic strategy has been successful.
9. Figure 3K shows data in a model of unilateral ischemia reperfusion injury in combination with

hypercholesterolemia. If this is to be incorporated, then more detail is required here. What is the degree of renal injury and quantification should be provided?

10. The authors make a case that arterial markers are lost in their experimental model. Ideally, kidney endothelial cells should be assessed rather than the whole kidney. Quantification needs to be provided on the localisation of PNA^d, used as a marker of high endothelial venules. Again, whole kidney analysis are used to assess mRNA levels of other markers in Figure 4D.

11. To complete the story, it would be of interest if the authors could find any evidence of reduced Notch signalling in any of the diseases, they outline which are associated with tertiary lymphoid structures such as lupus, glomerulonephritis, IgA-nephritis or kidney transplants.

Reviewer #3 (Remarks to the Author):

Summary:

The authors used mouse genetics to investigate the role of Rbpj - a key effector of canonical Notch signaling - in adult renal endothelium. They observe spontaneous formation of tertiary lymphoid structures (TLSs) adjacent to second and third-order arteries upon loss of Notch signaling. They further characterize these TLSs by analyzing gene expression, cellular composition, and overall architecture. Utilizing different mouse models, they rule out the confounding effects of lymphatics and heart failure. Mechanistically, they show that a shift from an arterial to a high endothelial phenotype contributes to TLS formation in Rbpj mutant mice. The authors conclude that endothelial Notch signaling is critical for TLS formation.

General comment:

The present work identifies blood vessels, particularly arteries, as a potential driver of TLS formation in adult kidney. Overall, the work is intriguing and of scientific interest. Yet, some concerns need to be addressed.

Major points:

1. The authors argue that the arterial Notch signaling plays a crucial role in TLS formation, given the close spatial relationship between TLS and renal artery. It would be more convincing to use an arterial-specific Cre line, such as the Bmx-CreERT2 deleter, to knock out Rbpj and investigate the resulting phenotypes.
2. Since TLS can form in various organs such as lung and liver, under different conditions (Pipi et al. 2018), one is curious if there is TLS formation in other organs?
3. The phenotype shift in Rbpj-deficient ECs of the kidney is shown by expression changes of defined markers. How the loss of Rbpj leads to such a shift is not studied. RNA-sequencing of kidney endothelial cells might provide clues about underlying mechanisms.

Minor points:

1. Line 98: the authors examine the expression change of Hey1. Why not directly check the expression of Rbpj?
2. Fig 1 E vs. Fig 3 E, clearing and staining protocol seem inconsistent.

References

1. Pipi et al. Tertiary lymphoid structures: autoimmunity goes local. *Front Immunol.* 2018
2. Kusumbe et al. Age-dependent modulation of vascular niches for hematopoietic stem cells. *Nature* 2016

RESPONSE TO REVIEWERS' COMMENTS

We would like to thank the reviewers for all their constructive and insightful comments and take to opportunity to respond in detail. We apologize for the delay of the revised manuscript due to Corona restrictions.

REVIEWER COMMENTS

Reviewer #1 (Remarks to the Author):

This manuscript addresses an important issue as to the mechanisms underlying the formation of TLS. The authors show that endothelial cell-specific deletion of *rbpj* leads to TLS in murine kidneys. The kidney TLS are located around arteries. The demonstration that the *cdh5* notch signaling pathway is involved in TLS formation is interesting and potentially important.

However, there are several major issues with the body of evidence presented that need to be addressed.

1. *Cdh5*-specific deletion of *rbpj* is an EC phenotype most likely in all endothelial cells of all organs including the brain. The authors observe, however, TLS specifically in the kidney: No mechanism is presented or discussed why TLS form in the kidney.

It is correct that in our *Cdh5*-specific deletion, we have a general endothelial knockout and therefore would expect the phenotype not exclusively in the kidney. Thank you very much for this very relevant suggestion. We have now investigated other major organs: heart, liver and lungs. *Rbpj*-mutant mice show TLS formation in all investigated livers and lungs, but not in the heart. This corroborates our findings in the kidney and points to a general role of vascular-endothelial Notch in TLS formation in parenchymatous organs. These new findings are now incorporated into Figure 1 and Suppl. Figure 2 and in the text on page 7, last paragraph. We have maintained the focus on the kidney for TLS characterization because of the pronounced phenotype and our own interest.

2. A comprehensive TLS search evaluation should be performed in other major organs including the heart, the liver, the lung and the brain and more to show that the observation is kidney-specific.

We agree to this suggestion and have performed a comprehensive TLS search in liver, lung, kidney and heart. We now report TLS in kidney, lung and liver, but not the heart. The new data have been added to Figure 1 and Suppl. Figure 2 and described in the text on p. 7.

We are aware of the very interesting data on cerebral TLS in a mouse model of systemic lupus erythematoses¹ that suggest that choroid plexus TLS may play a role in neuropsychiatric lupus manifestation. We agree that it would be interesting to examine the brains of our endothelial *Rbpj*-Knockout model at the time point when we find TLS in other vascular beds. However, to maintain the focus on parenchymatous organs and to keep the experiments manageable during the time of intermittent lock down we omitted brain analysis. This question requires further study, which our study will surely prompt. But even without the brain data, we can show that TLS spontaneously occur in different parenchymatous organs after endothelial *Rbpj*-KO.

3. The authors claim that secondary lymphoid organs remain unchanged but they do not provide comprehensive evidence for this claim. What do they mean by "unchanged"?

They do not provide evidence that SLOs are unchanged other than selective (biased) analyses.

A claim of this nature should be substantiated by genome wide expression analyses of lymph nodes and spleen preferably using single cell transcriptome analyses with and without *rbpj* deletion in ECs and immune cells.

This is a good point, and we would like to rephrase our wording and be more specific.

We mean to describe the following: While we see a significant increase in B and T lymphocyte frequencies in kidney and liver by flow cytometry (Fig. 1B, Suppl. Fig 1E, Suppl. Fig. 2F), we do not see a difference in B and T cell frequency in the secondary lymphatic organs spleen and lymph nodes (Fig. 1 B, Suppl. Fig 1E for b lymphocyte subsets and progenitors), neither do we have a significant change in organ size between groups (Suppl. Fig. 1G).

We analyzed B cell subsets and show that follicular B cells, not progenitors or other subsets, are responsible for the B cell number increase in the kidney (Fig. 1 C).

In kidney, liver, blood, bone marrow and spleen, we have analyzed not only lymphocytes, but also myeloid lineage cells using antibody panels that are established in our lab to characterize different myeloid subsets^{2, 3, 4, 5, 6}; while this is of course not as unbiased as single cell RNA sequencing would be, it is a method that we have well established and that we are confident in.

We see a small, but significant increase in dendritic cell frequency (CX3CR1+ and CX3CR1neg DC) in the spleen in *Rbpj*^{ΔEC} (see Suppl. Fig 1H for gating strategy), which parallels the findings in kidney and bone marrow (Fig. 1B), but no other change in the monocyte/macrophage lineage in terms of relative cell distribution or absolute cell number. Dendritic cells are necessary for TLS formation, and specific depletion of DC resulted in less TLS in a model of virus induced bronchus-associated lymphoid tissue⁷, so this finding strengthens our TLS observation. In immunostaining, we see no difference in size and distribution of follicles in spleens and lymph nodes (data not shown).

We added a new FACS panel to differentiate B cell subsets and precursors in bone marrow, kidney, spleen and lymph nodes; we see a slight decrease in marginal zone b cells in *Rbpj*^{ΔEC} spleens, but no change in follicular b cell number in spleen or lymph node (Suppl. Fig 1E).

We have tried to also identify marginal zone b cells in histological sections via CD21/35 and B220 costaining, but failed to get good specificity due to the high B220 background within the structures as well as the close proximity of follicular b cells to CD21/35 positive FDC.

An exemplary FACS plot of lymph node Follicular B cells is shown here:

We have added the following description in the discussion on p. 17: *Our analysis also revealed that secondary lymphoid tissues, such as lymph nodes and spleen, were largely unaffected in mutant mice, judged by size and cell numbers. A small but significant increase in dendritic cell numbers was noted, which is consistent with TLS development, due to their role in TLS formation*⁷.

4. The authors observed 12 weeks after knockout higher frequencies of mature B cells in the kidney and lower frequencies in blood and the bone marrow and attribute these changes to a recruitment phenomenon into the kidney without evidence for this interpretation. There are many other possibilities to interpret these data, i.e. effects of the knockout in the bone marrow itself including its stem cell niche, immune cells and mesenchymal cells. The authors need to rule out these possibilities.

We agree with this point – there are other possibilities. In order to address this point, we have added multicolor flow cytometry experiments with a new antibody panel to differentiate B cell subsets (Suppl. Fig 1E, gating Suppl. Fig. 1D). If the observed phenotype was due to a change in the bone marrow stem cell niche, we would expect to see changes also on the B cell progenitor levels. However, neither in the bone marrow, nor in the periphery, are significant changes in B cell progenitors.

We also added bone marrow transcriptional analyses for Il7 and Cxcl12, key regulators of B cell development and migration, which are expressed in stromal cells in the bone marrow niche^{8,9,10}. In QRT-PCR, we do not observe changes in Il7 or Cxcl12 mRNA expression in the bone marrow between groups (Suppl. Fig. 1F). Thus, key chemokine components of the bone marrow niche are unaltered. Lastly, the B lymphocyte subset responsible for the increase in the kidney are follicular B cells, not progenitors or other forms, which also strengthens our point. The absence of changes in the bone marrow niche together with the presence of follicular B cells in peripheral TLS suggest recruitment as a reasonable explanation.

5. The authors show that there is an increase in dendritic cells in the kidney, bone marrow and spleen. These data appear to contradict their claim that the immune system only responds to Cdh5 rbpj signaling deficiency to recruit immune cells into kidney arteries but rather appears to indicate that the knockout is associated with major changes in the immune system itself possibly affecting the bone marrow and SLOs.

The changes observed in dendritic cells are consistent with development of TLS, since dendritic cells are necessary for TLS development, and an increase is expected⁷.

In a mouse model of bronchus-associated lymphoid tissue (BALT), the development of these lung TLS was impaired after CD11c-targeted knock-out of dendritic cells⁷. We agree that our wording oversimplified the interrelations and have tried to address this with more careful explanations in the text (page 6, last paragraph: *“Rbpj^{ΔEC} mice also showed an increased frequency of dendritic cells in the kidney, bone marrow and spleen (Fig. 1B, lower panel; Supplementary fig. 1H). In contrast, we observed no difference in neutrophilic granulocytes, monocyte subsets or macrophages between Rbpj^{ΔEC} and control mice, and no signs of overt systemic inflammation (Fig. 1B, lower panel, supplementary Fig. 1H). Together, these findings suggest active lymphocyte recruitment to the kidney.”*), as well as more detailed analyses as explained above and in the Supplementary Figure 1. The point about SLO and bone marrow we addressed above as well, we would expect stronger differences in SLO and BM size/volume and/or cell composition if this was the major driver of TLS development in our model. We did not observe such changes.

6. The authors perform selective transcript analyses in HEC.

These analyses are highly biased and are not sufficient to provide comprehensive and sufficient information on the phenotypes of the HEC in response to *rbpj* deletion in endothelial cells. The authors should perform single cell transcript analyses of sorted endothelial cells in major organs including the kidney with and without knockout to get a more complete picture of the phenotype of the various EC subtypes as expressed for example in t-SNE projections.

This is a misunderstanding. We did not selectively analyze HEC mRNA, as we did not find a way to isolate HEC from kidneys with sufficient yield; we likely lost membrane bound epitopes during the digestion and tissue disruption protocol. Instead, the presented analyses were from whole kidney mRNA, reflecting a pattern of loss of arterial markers which we chose according to¹¹. While Notch signaling is required for arterio-venous differentiation in development¹², it has previously not been shown that low level endothelial Notch signaling is also required for maintenance of arterial phenotype in the adult mouse, and that genetic *Rbpj* deletion induced in the adult animal (after completion of vasculo- and angiogenesis) leads to loss of arterial EC signature.

To corroborate and extend our findings on EC-specific gene expression changes, we cell-sorted EC from kidneys of littermate control and mutant mice. We performed unbiased transcript analyses, but also interrogated the transcripts with described signature gene sets derived from single cell analysis of kidney EC subpopulations. This analysis corroborated a striking loss of arterial signature in mutant EC across different arterial subpopulations analyzed. In addition, comparison with a published signature of high EC¹³ revealed a shift towards a homeostatic high EC signature on the transcriptome level, which was further corroborated by PNA staining in *Rbpj*^{ΔEC} on the protein level. We have added these new experiments and their description into the new Figure 5, described on p. 12ff.

While we found the idea of single cell transcript analysis very appealing, we have decided to run bulk RNA sequencing from sorted endothelial cells instead for the following considerations:

- Changes in Notch signaling often occur at a low (transcript) level and may be subtle; with limited sequencing depth, we might miss these changes in low read transcripts. As we wanted to see changes in the transcriptional signature, we chose deep bulk sequencing instead.
- Cell subtype assignment in single cell analyses is according to a certain set of marker genes. If we assume that our knockout leads to a change in gene signature or EC subtype specification, we might end up assigning the “wrong” cell subset or fail to see a shift in the signature.

Reviewer #2 (Remarks to the Author):

Fleig and colleagues present an interesting set of data supporting a role for endothelial Notch signalling in the formation of tertiary lymphoid structures in the kidney. The authors show that loss of the Notch signaling mediator Rbpj in endothelial cells, but not lymphatics, leads to tertiary lymphoid structures around renal arteries. The authors postulate that this phenotype involves a loss of arterial specification and acquisition of a "high endothelial cell" phenotype. The work presented is novel and an important finding, tertiary lymphoid structures are found in several renal diseases, but the molecular mechanisms are completely unknown. This paper begins to address this issue. Although of interest, there are several areas where the paper could be strengthened, in particular details about the deletion of Rbpj in the kidney endothelium, the description of histological changes in the kidney and the strategy used to argue that cardiac problems do not contribute to the renal phenotype as outlined below.

1. The authors take the strategy of deleting Rbpj using a Cdh5 promoter. However, there is considerable heterogeneity in the endothelium of the adult kidney. Do the authors have any insights into the specific vascular beds where Cdh5 is expressed in the normal adult kidney? Are they targeting large arteries, glomerular capillaries, peritubular capillaries, vasa rectae, lymphatics?

We would like to thank the reviewer for this very valid point. We apologize for not having included these controls in the first version of the manuscript. Cdh5 is expressed in all blood and lymphatic endothelial cells in mouse kidney. The inducible Cdh5-cre^{ERT2} we used¹⁴ from Ralf Adams' lab is used in many high impact publications (e.g.,^{15, 16, 17, 18}, and is with 349 references the most widely used Cdh5-Cre model (<http://www.informatics.jax.org/recombinase/summary?driver=Cdh5> ; <http://www.informatics.jax.org/reference/allele/MGI:3848982?typeFilter=Literature>) and within the EC field known for its great recombination efficiency and low background or leakiness (as opposed to e.g.¹⁹).

In the kidney, it is targeting EC of arteries, glomerular capillaries, peritubular capillaries, vasa rectae and lymphatics as well. In order to visualize this, we have added exemplary pictures of a reporter line (Cdh5Cre^{ERT2};TdTomato) to Suppl. Fig. 1 B to show specific endothelial expression of Cdh5 in all EC beds and the successful recombination strategy.

2. Limited data is provided to support the successful deletion of Rbpj in the kidney endothelium. Currently, only a qRT-PCR of Hey1 on the whole kidney is provided. Can information ideally on isolated endothelial cells examining Rbpj expression levels be provided? There is also little information in the literature on the expression of Rbpj and Hey1 in adult kidneys.

Rbpj is a DNA-bound transcription factor that represses transcription unless Notch receptor intracellular domain (NICD) is bound, which leads to activation of Notch target genes. As transcription of Notch target genes induced by all four receptors depends on the presence of Rbpj, deletion of Rbpj has been widely accepted as a model of inhibition of canonical Notch signaling^{6, 15, 17}. Conditional deletion of Rbpj by Cdh5-Cre transgene approach in kidney EC has also been demonstrated²⁰. Furthermore, the identical strategy for Cre transgene-dependent deletion of Rbpj in endothelial cells was used in numerous high-impact publications^{21, 22, 23}.

In order to monitor our recombination strategy in the kidney, we have chosen these three approaches:

1) Cre-recombination control via fluorescent reporter strains: Because we intended to study the effect on EC Notch knockout after completion of angiogenesis, we used the tamoxifen-inducible Cre-line *Tg(Cdh5-cre/ERT2)1Rha*¹⁴ which has a very high recombination efficiency when crossed and analysed with a fluorescent Cre-reporter (Suppl. Fig 1B). With the TdTomato-reporter, recombination is >98% and strictly endothelial.

2) Rbpj-recombination control via recombination-PCR from kidney lysate DNA. From whole kidney DNA, PCR specific for the recombination for the *Rbpj* locus after loxP-site-excision shows a band with the expected size (Suppl. Fig. 1A). Together with the information of EC-specific targeting by Cre, this confirms targeting of *Rbpj* in EC.

3) Analysis of specific Notch target genes by qPCR and RNAseq: *Hey1*, a specific Notch downstream target in EC²⁴, is significantly downregulated in whole kidney mRNA. Furthermore, in RNAseq analysis from sorted kidney EC, *Hey1* is significantly downregulated in *Rbpj*^{ΔEC} with an expression of 59% as compared to control with a p<0.05. Together, these results show consistent downregulation of Notch-dependent transcription in EC in our mutant mice.

We had not included Rbpj expression levels, because in our hands, several different primer pairs designed for Rbpj did not result in a reliable QPCR product from kidney or spleen mRNA. Given the specific targeting in EC, and the presence of Rbpj/ active Notch signaling in other kidney cell types, Western blots analysis for Rbpj protein expression is not suitable, because we do not have enough isolated EC to isolate enough protein for the analysis. Lastly, we also did not have a good Rbpj antibody for immunohistochemistry.

Can the authors provide any information - if Rbpj levels are already low in adult kidneys, then what is the rationale for dampening signalling further?

Notch signaling is important for vasculo- and angiogenesis²⁴ in general, but specifically for arterial identity of endothelial cells^{12,25}. In kidney vasculature, Notch is required for normal glomerular capillary EC maturation²⁶.

Single-cell RNAseq studies of coronary artery development show that coronary arteries develop from venous EC and that coronary arterial EC upregulate Notch1, Dll4 and Hey1 upon phenotype switch to arterial EC¹¹. So endothelial Notch signaling intensity is stronger in arteries and weaker in capillaries and veins. Rbpj deletion is widely accepted as a model for blockade of canonical Notch signaling^{6,15,17}, as all Notch receptor intracellular domains require DNA-bound Rbpj to initiate transcription of target genes. Therefore, the primary effect of Rbpj deletion is expected to show in Notch signaling intensity, particular in tissues with higher Notch signaling intensity, such as arteries, while little effect is expected in low-Notch tissues, such as veins. Our findings on various levels, best exemplified in our new transcriptional analysis, demonstrate exactly this effect, a downregulation of arterial signature (Fig. 5 C), thus strengthening also a functional role for Rbpj/Notch in adult arterial endothelial cells.

3. The authors present histological observations indicating the presence of tertiary lymphoid structures in the mice with deletion of endothelial Rbpj. It would be good to have a bit more information here, how many of these structures are seen in each mouse and where are they localised in the kidney (cortex/medulla, close to glomeruli). The lymphoid structures are described as being around arteries, but this is not clear to the reviewer in some of the pictures presented (e.g. mutant samples in Figure 1C, top panel).

Following the advice we have provided a schematic and a better representation of the anatomical location of the representative sections with magnifications in Fig. 1D. The lymphoid structures are localized predominantly around segmental and interlobal renal artery segments, no structures were found around small arterioles or glomeruli in our model without additional inflammatory stimuli. Furthermore, in Fig. 1F, 3D light sheet microscopy of blood vessels (red) and b lymphocytes (green) shows the same findings on a whole organ level.

In detail: In the top panel, the periarterial loose interstitial tissue seen in control animals (left side) is completely filled with mononuclear cells and obscuring a proper adventitia within these TLS. The lumen of the affected artery is presented in the close up as well. In addition, TLS in the hilar region are often found in a capsule fold in between the papilla/urinary space and the vascular hilus (e.g. Fig. 1D, the C-shaped area in the lower panel, lower magnification / second picture from the right).

In the lung, we see bronchus-associated lymphoid tissue in close proximity to bronchus and arterial structures. In the liver, we see lymphoid aggregates in periportal and pericentral regions – with the portal vein anatomically being accompanied by a small artery and bile duct (Fig. 1G).

How is the size of tertiary lymphatic structures being assessed, what is examined in a control animal?

In order to quantify, we used PAS stainings and measured the area filled with densely packed mononuclear cells, per section, within the periarterial vessel nerve sheath, which usually consists of loose interstitial tissue.

We have added this paragraph to the methods section, p. 27f:

“Kidneys were cut in half along the axial plane and mounted face down, resulting in sections representing the middle of the kidney with hilar, papillar, medullary and cortical fractions. In PAS stained sections, area covered by mononuclear cells was measured in the periarterial loose interstitial tissue area (vessel nerve sheath) along all (typically longitudinally cut) segmental and all (typically cross sectioned) interlobar arteries per section. As the kidneys/sections were of similar size, we did not normalize the measured area to the section size.”

In the control animals, area covered by mononuclear cells in the periarterial loose interstitial tissue is measured (single mononuclear cells were almost always found.) In addition, our data with flow cytometry provide an accurate and unbiased measure of the lymphoid cell burden (Fig. 1B, Suppl. Fig. 1 E), which corroborates the findings across all investigated mice. Furthermore, the size and location can also be appreciated in the representative light sheet microscopy images in Fig. 1F.

4. A degree of renal injury is indicated in the endothelial knock-out of Rbpj. Again, it would be optimal to see some more histological evidence to support this. Is the structure of the glomerulus altered or are there any signs of tubular injury? How about more specific functional markers such as albuminuria, creatinine clearance or blood urea nitrogen.

This is a very good point. We were analyzing the baseline kidney phenotype at 3 months after knockout induction. There is a sub-clinical degree of renal injury with more fibrosis staining in TLS areas and a small (2-fold) increase in KIM-1 mRNA (Suppl. Fig. 1A, B, C, D). We see no difference between groups in KIM-1 staining in the tubules (not shown). With albuminuria, blood and urine creatinine or blood urea nitrogen we see a large distribution range and no significant change between groups. We have added 6 and 12 week data below for your information.

5. Is the presence of lymphoid tertiary structures a renal-specific effect or is there any evidence of similar structures in other non-lymphoid organs?

To address this very valid point, we have analyzed liver, lung and heart in these mice. We see TLS in liver and lung, but not in the heart. We have added this data to the manuscript (Fig. 1G, Suppl. Fig. 2E, F, G) and text on p. 7, last paragraph.

6. The authors argue that cardiac failure may contribute to the development of tertiary lymphoid structures in their mice. However, it looks like cardiac failure occurs later than the time-points where the kidney study has been performed. What do the hearts look like at earlier time-points?

At the time of sacrifice, *Rbpj*^{ΔEC} mice do show mild signs of heart failure, e.g. an increase in heart weight/femur length ratio. The heart phenotype has also been described previously by us (data not reported) and by others²⁷. We have analyzed the heart and kidneys at earlier timepoints; at 6 weeks post induction, we do not see cardiac failure or TLS in mutant kidneys. Graphs of heart weight/femur length (n=7 per group at 6 wk, 20 per group at 12wk timepoint) and TLO area per kidney section (n=5-6 per group for 6wk, n=10 per group for 12wk timepoint) below; 2-way ANOVA with multiple comparisons; **p<0.01, ***p<0.001.

We therefore chose a genetically distinct cardiac failure model to find out if the TLS are a general secondary feature of cardiac failure in mice. We report that this is generally not the case.

Instead, a different strategy examining renal tertiary lymphoid formation in mice lacking Stat3 in the myocardium who develop heart failure is taken. No lymphoid structure formation in the kidney is found but it is difficult to see how the authors can really relate this different model to the observations they see in mice lacking endothelial *Rbpj*.

Does it really rule out cardiac failure driving the changes seen in renal tertiary lymphoid structures in their mice?

A valid but difficult point. We can at least say that TLS is not a general consequence of heart failure. This does of course not rule out that cardiac failure promotes TLS formation in the context of endothelial Notch loss of function. We have added a point to the discussion on p. 17: *Furthermore, although heart failure is associated with inflammatory changes, development of TLS was not a general feature of heart failure, since a genetically distinct heart failure model without alteration of Notch signaling in EC (data not show) did not show TLS. However, in the context of endothelial Notch loss-of-function, heart failure may contribute to development of TLS.*

Of note, one aim of our paper is to show that in the *Vecad;Rbpj* mouse model, used in many prior publications, multiple disease processes are present at the same time. One major change in different vascular beds in this model has so far not been described, and this is the TLS phenotype.

7. In Figure 3E, pictures of lymphatic structures are shown. If possible, the findings would be strengthened if the authors could show a movie of their data here.

We have added a movie to the online supplements from the dataset used for the 3D-rendering (Supplemental Movie 2).

8. The authors go on to show that renal tertiary lymphoid structures do not form in mice lacking *Rbpj* in *Prox1* expressing cells.

The authors should note some caveats here as *Prox1* is not solely expressed in lymphatics in the kidney (Kim YM, PLoS One. 2015; 10(5):e0127429; Kenig-Kozlovsky Y et al J Am Soc Nephrol. 2018 Apr; 29(4):1097-1107). As highlighted above, data should be provided to show the transgenic strategy has been successful.

Thank you for pointing out these two important articles on *Prox1* in kidney development. In contrast to the developing mouse kidney, in the adult mouse, *Prox1* is only expressed in kidney lymphatics (and ascending vasa recta / lymphatic and blood hybrid vessels).²⁸ We use the inducible *Prox1Cre^{ERT2}* from Taija Mäkinen's lab. To verify specificity, we have performed additional recombination control experiments and added the data in Suppl. Fig. 3, also see below.

We crossed the *Prox1Cre^{ERT2}* with the mTmG reporter line and induced recombination at 8 weeks of age (adult mouse), consistent with our experimental protocol. Suppl. Fig. 3 (also below) shows co-staining in paraffin-embedded sections for GFP (recombination) and Lyve1 (lymphatics). Within the kidney, we did not find venous valves and no apparent non-lymphatic EC targeting within the cortex and outer medulla region; the GFP+ cells are mostly also Lyve1+, in some areas (smaller lymphatic

vessels) not Lyve1-positive, but in shape and location clearly identifiable as lymphatic vessels (irregular lumen, periarterial location). While we did not notice tubular Prox1 expression in the *Prox1Cre^{ERT2};mTmG*-model induced at 8 weeks of age, in the inner medulla/papilla region, there was scarce recombination in cells that are not LYVE1-positive. These likely represent ascending vasa recta that have been shown to be hybrid specialized vessels with lymphatic (Prox1+, VEGFR3+) and blood endothelial characteristics²⁹. Thus, our model targets largely adult lymphatic endothelial cells in the kidney. On the other hand, since we do not have a phenotype in these mice, off-target effects, even if possible, are not a strong confounder of data.

9. Figure 3K shows data in a model of unilateral ischemia reperfusion injury in combination with hypercholesterolemia. If this is to be incorporated, then more detail is required here. What is the degree of renal injury and quantification should be provided?

We have summarized these additional data in Supplemental Figure 4, also shown below. The model was mainly used to illustrate that TLS in inflammatory models of kidney injury appear in similar locations and have the same morphology as those seen in our *Rbpj^{ΔEC}* model without an additional inflammatory stimulus. Ischemia reperfusion injury is a strong inflammatory stimulus in the affected kidney. The McMahon lab showed a B cell signature late in IRI repair³⁰, with a significant increase already at 3 months past IRI.

We used a model with 11 weeks follow-up post unilateral IRI, which leads to significantly decreased kidney size and weight (A, B).

Serum creatinine was unaltered, which is expected since the unaffected kidney compensates kidney function. In all I/R-kidneys, but not the contralateral kidney or Sham control kidneys, TLS form in a periarterial location and with comparable morphology as in our *Rbpj^{ΔEC}* model (D).

B, mm scale on left side.

D, Black bar, 1mm. White bar, 100μm. White arrow on HE pictures indicating artery.

10. The authors make a case that arterial markers are lost in their experimental model. Ideally, kidney endothelial cells should be assessed rather than the whole kidney.

This is an excellent point. We have now added the following experiment: we isolated kidney endothelial cells via MACS-preenrichment and FACS-sorting, and isolated RNA. We confirmed presence of TLO in kidney histology in ½ kidney of each mouse that we kept for this purpose. We then did bulk EC RNA deep sequencing. We compared the expression patterns of EC from *Rbpj^{AEC}* and control mice in an unbiased way and also performed gene set enrichment analysis compared to published gene sets of kidney EC subtypes derived from single cell analysis. This is summarized in Fig. 5 and described in detail on p. 12ff. On whole gene expression level, endothelial cell differentiation gene signatures, matrix organization and EC barrier function along with Notch signaling signatures were significantly downregulated in mutant EC. Furthermore, in GSEA with defined renal arterial cell transcriptomic signatures, significant and consistent downregulation was registered in each arterial transcriptomic signature from kidney - i. e. large artery, cortical artery, cortical arteriole, medullary arteriole, arteriole efferent – in mutant EC. In contrast, GSEA with transcriptomic profiles of homeostatic high EC revealed significant enrichment in mutant EC. Together, these findings demonstrate that loss of renal endothelial Notch signaling induces an EC phenotype shift characterized by renal arterial dedifferentiation and HEC gene upregulation.

Quantification needs to be provided on the localisation of PNAd, used as a marker of high endothelial venules.

We tried PNAd staining in flow cytometry but were not successful, likely due to destruction of the antigen during the digestion and tissue disruption steps. We were successful with PNAd staining on embedded tissue sections, which is the standard approach often used.

We found quantification challenging, as the localization and intensity of staining is very specific. We decided to measure grayscale intensity across endothelial cell borders, measuring 10 stretches of 10-12µm in at least two kidney sections per mouse across EC linings. In order to weigh peaks over background, we performed root mean square analysis; n=3 per group, **p<0.01. The quantification was added to the ms as Supplementary Fig. 4E.

Again, whole kidney analysis are used to assess mRNA levels of other markers in Figure 4D.

This is again a good point, please see response above for EC transcriptomic analysis, particularly the partial acquisition of high EC phenotype. In addition, these data demonstrate that expression of cell adhesion and trafficking molecules involved in leukocyte recruitment are upregulated on the whole organ RNA levels. In response we analyzed our EC transcriptomic expression data but could not

detect significant changes in mutant endothelial cells (see table below). This strongly suggests that upregulation of these cell adhesion molecules occurs in recruited leukocytes or reticular cells derived from the stromal cell compartment, which support lymphocyte recruitment, as described³¹. We have added a comment to this aspect into the ms discussion on p. 16.

Gene name	fold change KO	P-adj
Icam5	1.8276629	5.00E-04
Icam4	1.73547303	1.84E-03
Bcam	0.805275378	1.07E-01
Epcam	0.68423661	1.15E-01
Icam2	0.887512767	6.21E-01
Vcam1	0.911556098	9.54E-01
Icam1	1.031841413	9.54E-01
Madcam1	0.982570852	9.54E-01
Glycam1	0.973072828	9.54E-01
Sele	0.960065845	9.54E-01
Sell	0.957571808	9.54E-01
Selp	1.058993561	9.54E-01

11. To complete the story, it would be of interest if the authors could find any evidence of reduced Notch signalling in any of the diseases, they outline which are associated with tertiary lymphoid structures such as lupus, glomerulonephritis, IgA-nephritis or kidney transplants.

This is an excellent suggestion. The new data are presented in Fig. 5E and in the results part on p. 13. We have used a previously published dataset with RNAseq data from human kidneys with various diseases³². We have previously shown *Cxcl13* and *Ltb* to be upregulated in these diseases, indicative of TLS formation³³.

Across different chronic inflammatory kidney disease entities, there is statistically significant downregulation of Notch target genes, *Heyl* and *Hes1*, and to a lesser degree *Hey1*, in this dataset, indicating reduced Notch signaling in kidneys with chronic inflammation.

Reviewer #3 (Remarks to the Author):

Summary:

The authors used mouse genetics to investigate the role of *Rbpj* - a key effector of canonical Notch signaling - in adult renal endothelium. They observe spontaneous formation of tertiary lymphoid structures (TLSs) adjacent to second and third-order arteries upon loss of Notch signaling. They further characterize these TLSs by analyzing gene expression, cellular composition, and overall architecture. Utilizing different mouse models, they rule out the confounding effects of lymphatics and heart failure. Mechanistically, they show that a shift from an arterial to a high endothelial phenotype contributes to TLS formation in *Rbpj* mutant mice. The authors conclude that endothelial Notch signaling is critical for TLS formation.

General comment:

The present work identifies blood vessels, particularly arteries, as a potential driver of TLS formation in adult kidney. Overall, the work is intriguing and of scientific interest. Yet, some concerns need to be addressed.

Major points:

1. The authors argue that the arterial Notch signaling plays a crucial role in TLS formation, given the close spatial relationship between TLS and renal artery. It would be more convincing to use an arterial-specific Cre line, such as the *Bmx-CreERT2* deleter, to knock out *Rbpj* and investigate the resulting phenotypes.

We thank the reviewer for this very valid point. We repeated the experimental protocol with inducible *BmxCre^{ERT2};Rbpj^{fl/fl}*, which target arterial EC^{2, 15}.

There was no development of TLS in this model, neither on tissue sections, nor by flow cytometry. However, recombination efficiency after induction in adult mice (as per our protocol), assessed by *BmxCre^{ERT2};mTmG* reporter mice, was high in major large arteries, but less good in the smaller arteries of the kidney, with a clear gradient from proximal to distal. When we specifically analyzed interlobar arteries (where we find most TLS in the *Rbpj^{ΔEC}* mice), recombination was about 50-60% (see new Supplementary Fig. 7 E: upper panel, interlobar artery; lower panel, segmental artery).

While this suggests that loss of major arterial Notch signaling is not sufficient to induce TLS formation, it certainly does not rule out a role for smaller arteries or arterioles, or arterial-type capillaries. Due to technical limitations, the data unfortunately remain somewhat inconclusive. These data are described in Suppl. Fig. 7 and results p. 14.

2. Since TLS can form in various organs such as lung and liver, under different conditions (Pipi et al. 2018), one is curious if there is TLS formation in other organs?

This is an excellent point – a general EC knockout would be expected have a phenotype in several EC beds. We have performed a comprehensive TLS search in liver, lung, kidney and heart and now report TLS in kidney, lung and liver, but not the heart. The new data have been added to Figure 1 and Suppl. Figure 1D and described in the text on p. 7.

3. The phenotype shift in Rbpj-deficient ECs of the kidney is shown by expression changes of defined markers. How the loss of Rbpj leads to such a shift is not studied. RNA-sequencing of kidney endothelial cells might provide clues about underlying mechanisms.

Following the suggestion, we sorted kidney EC from both experimental groups and performed bulk RNA deep sequencing. We performed unbiased transcript analyses, but also interrogated the transcripts with described signature gene sets derived from single cell analysis of kidney EC subpopulations. This analysis not only corroborated a loss of Notch signaling in mutant EC, but also demonstrated a striking loss of arterial signature in mutant EC across different arterial subpopulations analyzed. In addition, comparison with published signatures of high EC¹³ revealed a shift towards a homeostatic high EC signature on the transcriptome level, which is consistent with the findings presented in histology. We have added these new experiments and their description into the new Figure 5, described in detail on p. 12ff.

This is the first description of a loss of endothelial-arterial signature following Notch inactivation in the adult kidney. The data are consistent with developmental data showing not only an association of Notch signaling intensity with arterial signature but even an active role of Notch signaling in arterial differentiation. At the same time, this also suggest mechanistically that maintenance of arterial signature requires active Notch signaling, and that a transition from high to low Notch signaling, which affects mostly arterial EC, is involved in acquisition of an high EC phenotype.

Minor points:

1. Line 98: the authors examine the expression change of Hey1. Why not directly check the expression of Rbpj?

Mainly – likely due to splice variants – we were not successful in getting a reliable QPCR signal and a single product on gel after Rbpj QRT-PCR using several different primer pairs.

On a more theoretical level, the loxP-sites in the Rbpj model are flanking the DNA binding site of Rbpj (exon 6+7). We are not sure if the deletion alters gene transcription; we think translation is where the knockout is “symptomatic”, as the DNA binding domain is missing. Also, Rbpj serves as a docking site for activated Notch intracellular domain, and its basic expression is generally low.

Hey1 is downstream canonical Notch signaling / downstream Rbpj in endothelial cells²⁴ and therefore appeared a good readout of canonical EC Notch signaling. In addition, our RNA seq data also demonstrate loss of Notch signaling (Fig. 5).

2. Fig 1 E vs. Fig 3 E, clearing and staining protocol seem inconsistent.

We apologize for not being more explicit, there is a technical explanation. The light sheet imaging was performed on two different microscopes (different laser equipment) with two different collaboration partners (see also main manuscript methods section, p. 28 last paragraph). The clearing and staining protocols are exactly the same, but the fluorophore wavelengths differ. In detail, the microscope used for figure 1 (Immunodynamics, UK Essen, Germany) has a very far red laser that can detect AF790 fluorescence. The higher the wavelength, the lower autofluorescence in the kidney; EC have been labeled with AF790 in Fig. 1E, and B cells with AF 647, therefore we have a nice and low background imaging in this staining.

For Fig. 3E, we used a different lightsheet microscope, which has no far-red laser and cannot detect AF790. We stained B220 with AF 647 (still low autofluorescence), but Prox1 and LYVE1 with Cy3, which unfortunately is in a range of high autofluorescence in the kidney in the channel used for Lyve1-staining. The other difference is that fig. 1E has been acquired in higher resolution (whole kidney, much larger size dataset) than Fig. 3E, where we have in high resolution only the region used for the (now new) short movie.

We have listed the two imaging protocols and microscopes in the methods section in the revised ms.

References

1. Stock AD, *et al.* Tertiary lymphoid structures in the choroid plexus in neuropsychiatric lupus. *JCI Insight* **4**, (2019).
2. Gamrekelashvili J, *et al.* Regulation of monocyte cell fate by blood vessels mediated by Notch signalling. *Nat Commun* **7**, 12597 (2016).
3. Gamrekelashvili J, *et al.* Notch and TLR signaling coordinate monocyte cell fate and inflammation. *Elife* **9**, (2020).
4. Getzin T, *et al.* The chemokine receptor CX3CR1 coordinates monocyte recruitment and endothelial regeneration after arterial injury. *EMBO Mol Med* **10**, 151-159 (2018).
5. Kapanadze T, *et al.* Multimodal and Multiscale Analysis Reveals Distinct Vascular, Metabolic and Inflammatory Components of the Tissue Response to Limb Ischemia. *Theranostics* **9**, 152-166 (2019).
6. Krishnasamy K, *et al.* Blood vessel control of macrophage maturation promotes arteriogenesis in ischemia. *Nat Commun* **8**, 952 (2017).
7. Halle S, *et al.* Induced bronchus-associated lymphoid tissue serves as a general priming site for T cells and is maintained by dendritic cells. *J Exp Med* **206**, 2593-2601 (2009).
8. Tikhonova AN, *et al.* The bone marrow microenvironment at single-cell resolution. *Nature* **569**, 222-228 (2019).

9. Egawa T, *et al.* The earliest stages of B cell development require a chemokine stromal cell-derived factor/pre-B cell growth-stimulating factor. *Immunity* **15**, 323-334 (2001).
10. Peschon JJ, *et al.* Early lymphocyte expansion is severely impaired in interleukin 7 receptor-deficient mice. *J Exp Med* **180**, 1955-1960 (1994).
11. Su T, *et al.* Single-cell analysis of early progenitor cells that build coronary arteries. *Nature* **559**, 356-362 (2018).
12. Lawson ND, *et al.* Notch signaling is required for arterial-venous differentiation during embryonic vascular development. *Development* **128**, 3675-3683 (2001).
13. Veerman K, Tardiveau C, Martins F, Coudert J, Girard JP. Single-Cell Analysis Reveals Heterogeneity of High Endothelial Venules and Different Regulation of Genes Controlling Lymphocyte Entry to Lymph Nodes. *Cell Rep* **26**, 3116-3131 e3115 (2019).
14. Wang Y, *et al.* Ephrin-B2 controls VEGF-induced angiogenesis and lymphangiogenesis. *Nature* **465**, 483-486 (2010).
15. Ehling M, Adams S, Benedito R, Adams RH. Notch controls retinal blood vessel maturation and quiescence. *Development* **140**, 3051-3061 (2013).
16. Luo W, *et al.* Arterialization requires the timely suppression of cell growth. *Nature* **589**, 437-441 (2021).
17. Pontes-Quero S, *et al.* High mitogenic stimulation arrests angiogenesis. *Nat Commun* **10**, 2016 (2019).
18. Tombor LS, *et al.* Single cell sequencing reveals endothelial plasticity with transient mesenchymal activation after myocardial infarction. *Nat Commun* **12**, 681 (2021).
19. Alva JA, *et al.* VE-Cadherin-Cre-recombinase transgenic mouse: a tool for lineage analysis and gene deletion in endothelial cells. *Dev Dyn* **235**, 759-767 (2006).
20. Wang Y, *et al.* Blocking Notch in endothelial cells prevents arteriovenous fistula failure despite CKD. *J Am Soc Nephrol* **25**, 773-783 (2014).
21. Benedito R, *et al.* The notch ligands Dll4 and Jagged1 have opposing effects on angiogenesis. *Cell* **137**, 1124-1135 (2009).
22. Ramasamy SK, Kusumbe AP, Wang L, Adams RH. Endothelial Notch activity promotes angiogenesis and osteogenesis in bone. *Nature* **507**, 376-380 (2014).

23. Kusumbe AP, Ramasamy SK, Adams RH. Coupling of angiogenesis and osteogenesis by a specific vessel subtype in bone. *Nature* **507**, 323-328 (2014).
24. Fischer A, Schumacher N, Maier M, Sendtner M, Gessler M. The Notch target genes Hey1 and Hey2 are required for embryonic vascular development. *Genes Dev* **18**, 901-911 (2004).
25. Hasan SS, *et al.* Endothelial Notch signalling limits angiogenesis via control of artery formation. *Nat Cell Biol* **19**, 928-940 (2017).
26. Farber G, *et al.* Glomerular endothelial cell maturation depends on ADAM10, a key regulator of Notch signaling. *Angiogenesis* **21**, 335-347 (2018).
27. Jabs M, *et al.* Inhibition of Endothelial Notch Signaling Impairs Fatty Acid Transport and Leads to Metabolic and Vascular Remodeling of the Adult Heart. *Circulation* **137**, 2592-2608 (2018).
28. Bazigou E, *et al.* Genes regulating lymphangiogenesis control venous valve formation and maintenance in mice. *J Clin Invest* **121**, 2984-2992 (2011).
29. Kenig-Kozlovsky Y, *et al.* Ascending Vasa Recta Are Angiopoietin/Tie2-Dependent Lymphatic-Like Vessels. *J Am Soc Nephrol* **29**, 1097-1107 (2018).
30. Cippa PE, Liu J, Sun B, Kumar S, Naesens M, McMahon AP. A late B lymphocyte action in dysfunctional tissue repair following kidney injury and transplantation. *Nat Commun* **10**, 1157 (2019).
31. Katakai T, *et al.* Organizer-like reticular stromal cell layer common to adult secondary lymphoid organs. *J Immunol* **181**, 6189-6200 (2008).
32. Grayson PC, *et al.* Metabolic pathways and immunometabolism in rare kidney diseases. *Ann Rheum Dis* **77**, 1226-1233 (2018).
33. Fleig SV, *et al.* Long-term B cell depletion associates with regeneration of kidney function. *Immun Inflamm Dis*, (2021).

Reviewer #1 (Remarks to the Author):

The revised manuscript NCOMMS-20-19528A has been considerably improved. Most of my concerns have either been addressed experimentally or explained by the authors. As stated in my original comments, the work provides important new insights into the formation of TLS formation in that it identifies an indeed previously unidentified role for vascular endothelial cells and Notch signaling in arterial TLS neogenesis. This finding is of general interest for the broad readership of Nature Communications.

Although the authors were asked to perform single cell transcriptome analyses, they explain in plausible terms that these could be performed in the future. This reader agrees with that suggestion and agrees that single cell EC transcriptome analyses are not absolutely required to arrive at the conclusions.

The authors claim that the endothelium and the Notch pathway is not involved in heart TLS formation. There are several caveats that the authors need to address before the manuscript becomes acceptable: It is possible that the detection of TLS in the heart escaped their analyses as the heart is a large organ and ruling out TLS in the heart is quite an immense experimental task. Moreover, TLS in the heart in the knockout mice could still develop at a later time window of the mice's lifetime as it may be related to aging. The authors should discuss this possibility briefly and concisely. Therefore, the authors should include a cautionary note regarding their negative finding for the heart and also state that given the new findings more work needs to be performed in the future to study further parenchymal tissues including the brain.

Andreas Habenicht

Reviewer #2 (Remarks to the Author):

This revised article presents the finding that loss of the Notch signaling mediator Rbpj in endothelial cells, but not lymphatics, leads to tertiary lymphoid structures around renal arteries. The findings are of interest and importance in the field. In the revised manuscript, the authors have responded well to the reviewer queries clarifying issues around Cre lines, histological analysis and details of the model of renal injury used in the study. Additional data is provided regarding tertiary lymphoid structures in other organs, bulk-sequencing in endothelial cells to support the conclusion that arterial markers are lost in their experimental model and relevance of their findings to renal patients.

Reviewer #3 (Remarks to the Author):

Fleig and colleagues have submitted a revised version of their manuscript which addresses most of my previous comments. Overall, the results are interesting and novel and will make an important contribution to the journal.

RESPONSE TO REVIEWERS' COMMENTS

We would like to thank the reviewers for all their constructive and insightful comments and again respond in detail here.

REVIEWER COMMENTS

Reviewer #1 (Remarks to the Author):

The revised manuscript NCOMMS-20-19528A has been considerably improved. Most of my concerns have either been addressed experimentally or explained by the authors. As stated in my original comments, the work provides important new insights into the formation of TLS formation in that it identifies an indeed previously unidentified role for vascular endothelial cells and Notch signaling in arterial TLS neogenesis. This finding is of general interest for the broad readership of Nature Communications.

Although the authors were asked to perform single cell transcriptome analyses, they explain in plausible terms that these could be performed in the future. This reader agrees with that suggestion and agrees that single cell EC transcriptome analyses are not absolutely required to arrive at the conclusions.

The authors claim that the endothelium and the Notch pathway is not involved in heart TLS formation. There are several caveats that the authors need to address before the manuscript becomes acceptable: It is possible that the detection of TLS in the heart escaped their analyses as the heart is a large organ and ruling out TLS in the heart is quite an immense experimental task. Moreover, TLS in the heart in the knockout mice could still develop at a later time window of the mice's lifetime as it may be related to aging. The authors should discuss this possibility briefly and concisely. Therefore, the authors should include a cautionary note regarding their negative finding for the heart and also state that given the new findings more work needs to be performed in the future to study further parenchymal tissues including the brain.

Andreas Habenicht

We would like to thank the reviewer for this clarification. While our screening analysis did not find evidence for TLS in the heart, this certainly does not rule the possibility of TLS development in the heart. We have added a discussion on p. 16:

While our screening analysis of mutant mice did not find evidence for TLS formation in the heart, this certainly does not rule out the possibility of TLS formation in the heart, which might occur in structures not included in our analysis or in a different time frame.

Reviewer #2 (Remarks to the Author):

This revised article presents the finding that loss of the Notch signaling mediator Rbpj in endothelial cells, but not lymphatics, leads to tertiary lymphoid structures around renal arteries. The findings are of interest and importance in the field. In the revised manuscript, the authors have responded well to the reviewer queries clarifying issues around Cre lines, histological analysis and details of the model of renal injury used in the study. Additional data is provided regarding tertiary lymphoid structures in other organs, bulk-sequencing in endothelial cells to support the conclusion that arterial markers are lost in their experimental model and relevance of their findings to renal patients.

Thank you for your constructive comments!

Reviewer #3 (Remarks to the Author):

Fleig and colleagues have submitted a revised version of their manuscript which addresses most of my previous comments. Overall, the results are interesting and novel and will make an important contribution to the journal.

Thank you for your constructive comments!